# Jag1/2 maintain esophageal homeostasis and suppress foregut tumorigenesis by restricting the basal progenitor cell pool

Haidi Huang[1,12], Yu Jiang[1,12], Jiangying Liu[1], Dan Luo[1], Jianghong Yuan[1], Rongzi Mu[1], Xiang Yu ®[1], Donglei Sun[1], Jihong Lin[2], Qiyue Chen[3,4], Xinjing Li[1], Ming Jiang[5], Jianming Xu ®[6], Bo Chu ®[7], Chengqian Yin ®[8], Lei Zhang[8,9], Youqiong Ye ®[10], Bo Cao[1], Qiong Wang ®[11] ✉ & Yongchun Zhang ®[1] ✉

Basal progenitor cells are crucial for maintaining foregut (the esophagus and forestomach) homeostasis. When their function is dysregulated, it can promote inflammation and tumorigenesis. However, the mechanisms underlying these processes remain largely unclear. Here, we employ genetic mouse models to reveal that Jag1/2 regulate esophageal homeostasis and foregut tumorigenesis by modulating the function of basal progenitor cells. Deletion of Jag1/2 in mice disrupts esophageal and forestomach epithelial homeostasis. Mechanistically, Jag1/2 deficiency impairs activation of Notch signaling, leading to reduced squamous epithelial differentiation and expansion of basal progenitor cells. Moreover, Jag1/2 deficiency exacerbates the deoxycholic acid (DCA)-induced squamous epithelial injury and accelerates the initiation of squamous cell carcinoma (SCC) in the forestomach. Importantly, expression levels of JAG1/2 are lower in the early stages of human esophageal squamous cell carcinoma (ESCC) carcinogenesis. Collectively, our study demonstrates that Jag1/2 are important for maintaining esophageal and forestomach homeostasis and the onset of foregut SCC.

The luminal surface of the adult esophagus is lined with a layer of squamous stratified epithelium, which forms a barrier against various kinds of damage[1–4]. The esophageal epithelium is maintained by a single layer of basal progenitor cells that express p63, Krt5, and Krt14, as well as adhesion molecules such as Itga6, Itgb1, and Itgb4 at the base[2,3,5–7]. These cells have the ability to undergo self-renewal to replenish the stem cell pool and differentiate into suprabasal cells expressing Krt4 and Krt13. The balance between self-renewal and

[1]State Key Laboratory of Microbial Metabolism, Joint International Research Laboratory of Metabolic and Developmental Sciences, School of Life Sciences and Biotechnology, Shanghai Jiao Tong University, Shanghai 200240, PR China. [2]Department of Thoracic Surgery, Fujian Medical University Union Hospital, Fuzhou 350001, PR China. [3]Department of Gastric Surgery, Fujian Medical University Union Hospital, Fuzhou 350001, PR China. [4]Department of General Surgery, Fujian Medical University Union Hospital, Fuzhou 350001, PR China. [5]Center for Genetic Medicine, the Fourth Affiliated Hospital, Zhejiang University School of Medicine, Hangzhou 310030 Zhejiang, PR China. [6]Department of Molecular and Cellular Biology, Baylor College of Medicine, Houston, TX 77030, USA. [7]Department of Cell Biology, School of Basic Medical Sciences, Cheeloo College of Medicine, Shandong University, Jinan 250012 Shandong, PR China. [8]Institute of Cancer Research, Shenzhen Bay Laboratory, Shenzhen 518107 Guangdong, PR China. [9]State Key Laboratory of Chemical Oncogenomics, School of Chemical Biology and Biotechnology, Peking University Shenzhen Graduate School, Shenzhen 518055 Guangdong, PR China. [10]Department of Immunology and Microbiology, Shanghai Jiao Tong University School of Medicine, Shanghai 200025, PR China. [11]Department of Histoembryology, Genetics and Developmental Biology, Shanghai Key Laboratory of Reproductive Medicine, Key Laboratory of Cell Differentiation and Apoptosis of Chinese Ministry of Education, Shanghai Frontiers Science Center of Cellular Homeostasis and Human Diseases, Shanghai Jiao Tong University School of Medicine, Shanghai 200025, PR China. [12]These authors contributed equally: Haidi Huang, Yu Jiang. ✉e-mail: wangqiong@shsmu.edu.cn; yongchun_zhang@sjtu.edu.cn

differentiation is important for preserving esophageal homeostasis. Moreover, the basal progenitor cells are aligned approximately perpendicular to the basement membrane[2,3,8]. The alignment is maintained by cell junction molecules and apicobasal polarity and is essential for upholding the structural integrity of the esophageal epithelium. Over the past decades, genetic mouse models have served as a valuable tool for investigating the molecular mechanisms that regulate esophageal homeostasis, as the mouse esophagus has a similar structure to the human esophagus[2,3,9–11]. Notably, the forestomach epithelium of mice is also squamous stratified and is considered an expansion of the esophagus[12,13], making it a suitable model for studying foregut homeostasis and tumor initiation[14–16]. Despite recent advances in our understanding of squamous epithelial homeostasis in the esophagus and forestomach[17], our knowledge of the underlying molecular mechanisms remains limited.

Disruption of esophageal epithelial homeostasis can promote various esophageal diseases, such as epithelial injury and cancer. The esophageal squamous epithelium is often damaged by the reflux of bile acids[11,18], leading to the development of severe inflammation and basal cell hyperplasia[11,18–21]. Basal progenitor cell hyperplasia can progress to ESCC under the condition of chronic inflammation[22]. ESCC accounted for around 500,000 deaths worldwide in 2020[23]. It is characterized by highly proliferative basal progenitor cells with poor epithelial differentiation and strong inflammation[22,24–27]. Various types of immune cells, such as neutrophils, macrophages, and CD4+ and CD8+ T cells, are accumulating in the tumors[14,22]. While recent studies have revealed that the dysregulation of multiple pathways in squamous epithelial homeostasis can contribute to the initiation of esophageal or foregut SCC[14,28–30], the mechanisms are still not well understood.

Notch signaling is activated when ligands Jag1/2 and Dll1/3/4 on neighboring cells bind to cell membrane receptors Notch1-4[31,32]. This activation transactivates the transcription of many downstream targets, which are involved in various biological processes such as cell proliferation and differentiation[32–35]. In vitro studies have shown that Notch1/3 receptors are required for inducing esophageal epithelial differentiation[36]. Our recent studies utilized genetic mouse models to reveal that the loss of Jag1/2 reduces epithelial differentiation and squamous stratification during embryonic esophageal development[37]. These findings emphasize the significant contribution of Notch signaling in controlling the differentiation program in the esophagus. However, the role of Notch signaling in esophageal homeostasis at postnatal stages remains largely unclear. Investigating the function of Jag1/2 in esophageal homeostasis is therefore essential to addressing this knowledge gap.

Additionally, the role of Notch signaling in esophageal carcinogenesis has been a topic of controversy. Notch loss-of-function mutations are frequently observed in patients with ESCC[38,39], and Notch1 deletion and Notch inhibition have been shown to promote ESCC initiation in mouse models[30,40]. These results suggest that Notch signaling functions as a tumor suppressor in ESCC. However, other studies have shown that Notch1/3 overexpression in cancer cell lines promotes the development of ESCC[41], and increased expression of the Notch intracellular domain (NICD) is correlated with poor survival in patients with ESCC[42]. Furthermore, most of these studies have focused on the Notch receptors, and the role of Notch ligands such as Jag1/2 in SCC carcinogenesis remains unclear. Therefore, elucidating the function of Notch ligands will significantly contribute to our understanding of Notch singling in SCC initiation.

In this study, we demonstrate that Jag1/2 are essential for controlling the homeostasis of the esophageal and forestomach epithelium and suppressing the initiation of foregut SCC. Using genetic mouse models, we show that the deficiency of Jag1/2 disrupts esophageal and forestomach homeostasis. Through histological analyses, we uncover that Jag1/2 regulate the epithelial homeostasis by modulating the balance between self-renewal and differentiation of basal progenitor cells. Further mechanistic studies reveal that such balance is dysregulated due to impaired activation of Notch signaling upon Jag1/2 deletion. Additionally, Jag1/2 safeguard the squamous epithelium from injury induced by bile acids. More importantly, we find that the deletion of Jag1/2 accelerates foregut SCC initiation, and their expression levels are lower in the early stages of human ESCC carcinogenesis.

## Results
### Deletion of Jag1/2 leads to dysregulated homeostasis of the stratified squamous epithelium in the esophagus and forestomach
Since multiple ligands, including Jag1/2 and Dll1/3/4, can activate Notch signaling, we initially analyzed RNA sequencing data obtained from adult *wild type* (*WT*) mice to determine their expression levels in the esophageal epithelium. *Jag1/2* are expressed at higher levels compared to *Dll1/3/4* (Fig. 1a and Supplementary Fig. 1a). Immunostaining confirmed the high expression of Jag1 and Jag2 in the basal progenitor cells of the esophageal and forestomach epithelium (Fig. 1b, c and Supplementary Fig. 1b, c). The expression of these ligands in the human esophagus displays a similar pattern, with *JAG1* showing the highest expression (Supplementary Fig. 1d). Therefore, we focused on studying the role of Jag1/2 in the maintenance of the esophageal epithelium. To investigate this, we generated *p63^CreERT2/+*;*Jag1/2^loxp/loxp* mice in which *Jag1/2* were specifically deleted in the basal cells and their derivatives in the esophageal and forestomach epithelium upon tamoxifen administration (Fig. 1d and Supplementary Fig. 1e, f). The *Jag1/2 knockout* (*KO*) mice displayed disorganized structure in the esophageal epithelium with basal cell expansion and dilated intercellular space (Fig. 1e), as well as reduced esophageal epithelial thickness (Fig. 1f). The esophageal epithelium of *Jag1 KO* or *Jag2 KO* mice also displayed a similar phenotype, albeit with less severity (Supplementary Fig. 1g, h), suggesting a redundant role of these two ligands. Therefore, hereafter, our studies focused on the *Jag1/2 KO* mutants. It is noteworthy that the esophageal epithelium of *p63^CreERT2/+* mice retained the expression of p63 and remained normal (Supplementary Fig. 1i-k). Similarly, basal progenitor cell hyperplasia and intercellular space dilation were also observed in the forestomach squamous epithelium of *Jag1/2 KO* mice (Fig. 1g, h). Together, these results support that Jag1/2 are essential for maintaining the homeostasis of the stratified squamous epithelium in the esophagus and forestomach.

### Jag1/2 regulate multiple biological processes and signaling pathways
Next, to determine the mechanisms through which Jag1/2 regulate epithelial homeostasis, esophageal epithelia were isolated from *Jag1/2 KO* mice and *WT* mice as controls, and subjected to bulk RNA sequencing (Fig. 2a). A Pearson correlation coefficient analysis of the transcription profiles revealed consistent expression patterns among different samples in both *Jag1/2 KO* and *WT* mice (Fig. 2b), indicating high biological reproducibility. Notably, principal component analysis showed that *Jag1/2 KO* mice displayed a distinct expression profile compared to *WT* mice (Fig. 2c, d). Differential expression analysis identified 1141 and 1368 genes that were significantly upregulated and downregulated by at least 2 folds in the mutants, respectively (Fig. 2e). Gene Ontology enrichment analysis revealed that these genes regulate multiple biological processes, including cell division, mitotic cell cycle, keratinization, cell proliferation, keratinocyte differentiation, and cell-matrix adhesion (Fig. 2f), which are critical events in maintaining esophageal homeostasis[2,3,37]. Furthermore, KEGG pathway enrichment analysis showed that these genes are involved in various signaling pathways, such as pathways in cancer, DNA replication, cell cycle, cell adhesion molecules, basal cell carcinoma, and gap junction (Fig. 2g).

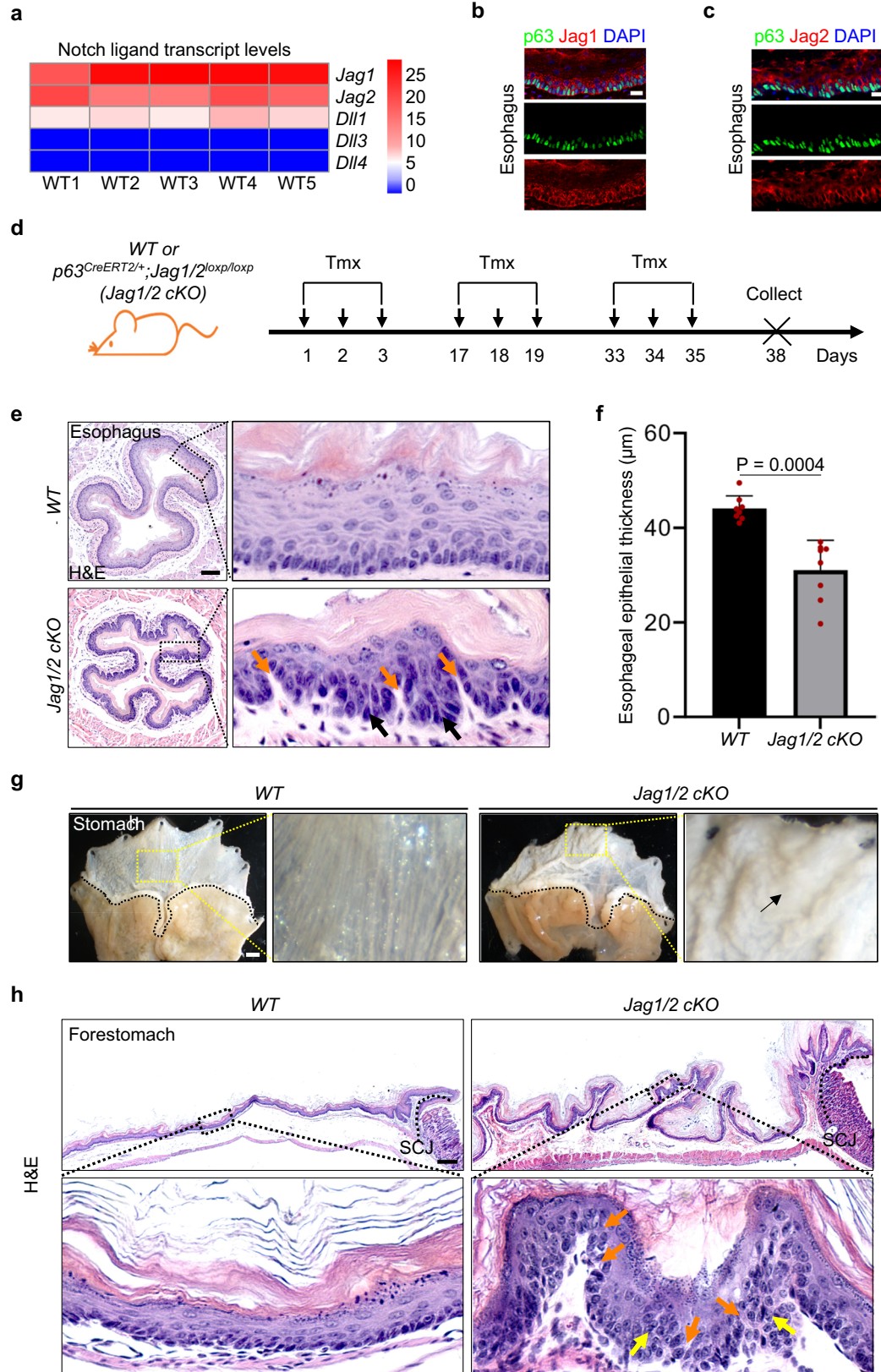

## The ablation of Jag1/2 dysregulates epithelial homeostasis by perturbing the balance between self-renewal and differentiation of basal progenitor cells

Given that the balance between self-renewal and differentiation of basal progenitor cells is critical for maintaining esophageal epithelial homeostasis[2] and altered cell proliferation and differentiation were

revealed by the RNA-seq analysis (Fig. 2f), our primary focus was to investigate how Jag1/2 influence this equilibrium. The transcript levels of *Ki67*, *Pcna*, and *Mcm2*, markers of cell proliferation, were significantly increased in *Jag1/2 KO* mutants (Fig. 3a), suggesting an increase in cell proliferation. The transcript levels of *Jag1/2* and the putative Notch downstream target *Hes1* were reduced in *Jag1/2 KO*

**Fig. 1 | Deletion of Jag1/2 leads to dysregulated homeostasis of the stratified squamous epithelium in the esophagus and forestomach. a** Transcript levels of Notch ligands *Jag1/2* and *Dll1/3/4* in the *wild type* (*WT*) mouse esophageal epithelium determined by RNA sequencing. Fragments Per Kilobase of transcript per Million mapped reads (FPKM) values are presented. **b, c** Immunofluorescence staining of p63, Jag1 (**b**) and Jag2 (**c**) in the esophageal epithelium of *WT* mice. Note the high expression of Jag1/2 in the basal cells, and the staining observed in the superficial layer is nonspecific marginal staining. Representative images are shown (n = 3 in (**b**), n = 3 in **c**). Scale bars: 20 μm. **d** Schematic diagram of the procedure for generating *Jag1/2* conditional knockout (*Jag1/2 cKO*) in the esophageal epithelium. *WT*, wild type; *Jag1/2 cKO, p63^{CreERT2/+};Jag1/2^{loxp/loxp}*. **e** Representative H&E stained esophageal sections. Note the increased basal cell number (darker blue nuclei, black arrows) and intercellular space (orange arrows) in the esophageal epithelium of *Jag1/2 cKO* mice (orange arrows). Scale bar: 100 μm. **f** The thickness of the esophageal epithelium is reduced in the *Jag1/2 cKO* mice. Data represent mean ± SD (n = 8 per genotype). *P* value was calculated by unpaired, two-tailed Student's t-test. **g** Representative gross morphology of the stomach. Note the hyperplasia in the forestomach of *Jag1/2 cKO* mice (black arrows). Scale bar: 2 mm. **h** Representative H&E stained sections of the *WT* and *Jag1/2 cKO* forestomach. Note the basal cell hyperplasia (yellow arrows) and increased intercellular space (orange arrows) in the forestomach epithelium of *Jag1/2 cKO* mice. SCJ, squamo-columnar junction. Representative images are shown (n = 6 per genotype). Scale bar: 200 μm.

mutants (Fig. 3a). Further immunostaining showed that p63[+] and p63[+]Ki67[+] basal progenitor cells are significantly increased in the *Jag1/2 KO* mice compared to *WT* mice (Fig. 3b, c), supporting that *Jag1/2 KO* promotes basal progenitor cell proliferation. On the contrary, the expression levels of suprabasal cell markers Krt4 and Krt13 were reduced, in contrast to the increase in basal cell markers p63 and Krt5 (Fig. 3d). Histological analysis showed that Krt4[+] suprabasal cells were reduced in the *Jag1/2 KO* esophageal epithelium (Fig. 3e, f), suggesting that Jag1/2 deletion decreases esophageal differentiation. Consistently, increased basal progenitor cell proliferation but reduced Krt4[+] suprabasal cells were observed in the forestomach epithelium of *Jag1/2 KO* mice (Fig. 3g–i). Collectively, these findings indicate that Jag1/2 ablation promotes basal progenitor cell proliferation while reducing epithelial differentiation in both the esophagus and forestomach, leading to deregulated homeostasis of the squamous epithelium.

## The absence of Jag1/2 results in elevated cell adhesion molecule expression, dilated intercellular space, and aberrant basal progenitor cell polarity

Our study revealed that Jag1/2 ablation results in basal cell hyperplasia, a condition frequently characterized by increased expression of basal cell adhesion molecules, dilated intercellular space, and altered cell polarities[43,44]. Consequently, we aimed to investigate whether these changes occur in the *Jag1/2 KO* mutants. In line with the expansion of basal progenitor cells, the transcript levels of basal cell molecules *Itga6*, *Itgb1*, and *Itgb4* were increased in *Jag1/2 KO* mice (Fig. 4a). Immunostaining verified an increase in the protein levels of Itga6, Itgb1, and Itgb4 in the basal layer of *Jag1/2 KO* esophageal and forestomach epithelium (Fig. 4b-d and Supplementary Fig. 2a-b). E-cadherin and Zo-1 immunostaining and transmission electron microscopy (TEM) revealed dilated intercellular space in the epithelium of the *Jag1/2 KO* esophagus and forestomach (Fig. 4e, f and Supplementary Fig. 2c-e), which is consistent with the results shown by H&E staining (Fig. 1e, h), and further shows the disruption of squamous epithelial integrity. We also observed that Jag1/2 deletion altered the expression pattern of PKCζ (Fig. 4g), which is normally expressed at the apical domains of p63[+] basal cells. Notably, the angles between the long axes of the p63[+] basal cell nuclei and the base membrane for the majority of basal progenitor cells in the *WT* epithelium are around 60-90 degrees (Fig. 4h). However, the angles in the *Jag1/2 KO* mutants were smaller and more randomly distributed (Fig. 4h). These results suggest dysregulation of basal cell polarities. Together, our findings showed that Jag1/2 regulate the expression of adhesion molecules, intercellular space, and basal cell polarities, which collectively contribute to the maintenance of esophageal and forestomach epithelial homeostasis.

## The deficiency of Jag1/2 reduces squamous epithelial differentiation through impaired Notch signaling

Given that Jag1/2 are important ligands for activating Notch signaling which has been shown essential for esophageal epithelial cell differentiation[36,37], we set out to determine whether the ablation of Jag1/2 impairs the Notch signaling pathway to reduce cell differentiation. We first conducted gene set enrichment analysis (GSEA) on the RNA-sequencing data comparing the esophageal epithelium of the *Jag1/2 cKO* mutants to the *control*. The analysis revealed a downregulation of Notch signaling in the *Jag1/2 cKO* mutants (Fig. 5a). Subsequent western blot analysis demonstrated a reduction in the protein levels of Notch1 intracellular domain (NICD1), the activated form of Notch1, upon Jag1/2 deletion (Fig. 5b). The expression of the differentiation marker Krt4 was decreased (Fig. 5c), in line with the immunostaining analysis (Fig. 3e, f). Furthermore, immunostaining showed that the nuclear Notch1[+] cells are reduced in the esophageal epithelium (Fig. 5d, e). These results demonstrated that Jag1/2 deletion impairs the activation of Notch signaling.

To further investigate whether Jag1/2 directly modulated epithelial cell differentiation, we generated 3D organoids from mouse esophageal epithelial cells (Fig. 5f). Deletion of Jag1/2 in the organoids led to the upregulation of the basal cell marker *p63* but a decrease in the expression of the suprabasal cell marker *Krt4* (Fig. 5g). Western blot analysis consistently showed downregulation of Krt4 protein levels (Fig. 5h). Moreover, immunostaining on organoids revealed an increase in p63[+] basal cells while showing a reduction in squamous differentiation with Jag1/2 deletion (Fig. 5i–k). Moreover, the protein levels of NICD1 were downregulated (Fig. 5h), and nuclear Notch1[+] cells were reduced in *Jag1/2 KO* organoids (Fig. 5l). These results together suggest that Jag1/2 within esophageal epithelial cells promote squamous differentiation through the Notch signaling pathway.

## Jag1/2 deficiency exacerbates squamous epithelial injury in the forestomach

Maintaining the integrity of the squamous epithelial barrier is vital in protecting tissues from a range of insults, including refluxates, thereby preventing the development of inflammation[45]. Since the integrity of the Jag1/2 squamous epithelium is disrupted, we set out to determine its impact on injury development. Mice were orally gavaged with bile acids, deoxycholic acid (DCA), a potent squamous epithelial damaging agent found in the refluxate[46,47] (Fig. 6a). The *Jag1/2 KO* mice exhibited significant weight loss compared to the *WT* mice (Fig. 6b) and mucosal hyperplasia in the forestomach (Fig. 6c). Histological analysis showed elevated squamous epithelial barrier damage, severe epithelial dysplasia, and inflammatory cell accumulation in the *Jag1/2 KO* forestomach (Fig. 6d-e). The compromised barrier resulted in increased permeability of the squamous epithelium, as demonstrated by Evans blue staining (Supplementary Fig. 3). Immunostaining showed that the number of proliferative basal progenitor cells (p63[+]Ki67[+]) was increased in the *Jag1/2 KO* mutants (Fig. 6f–g). Furthermore, immune cells, including Ly6G[+] neutrophils and Cd68[+] macrophages, were highly increased in the forestomach of *Jag1/2 KO* mutants (Fig. 6h-i). Taken together, these results demonstrate an important role of Jag1/2 in protecting mice from injury in the foregut.

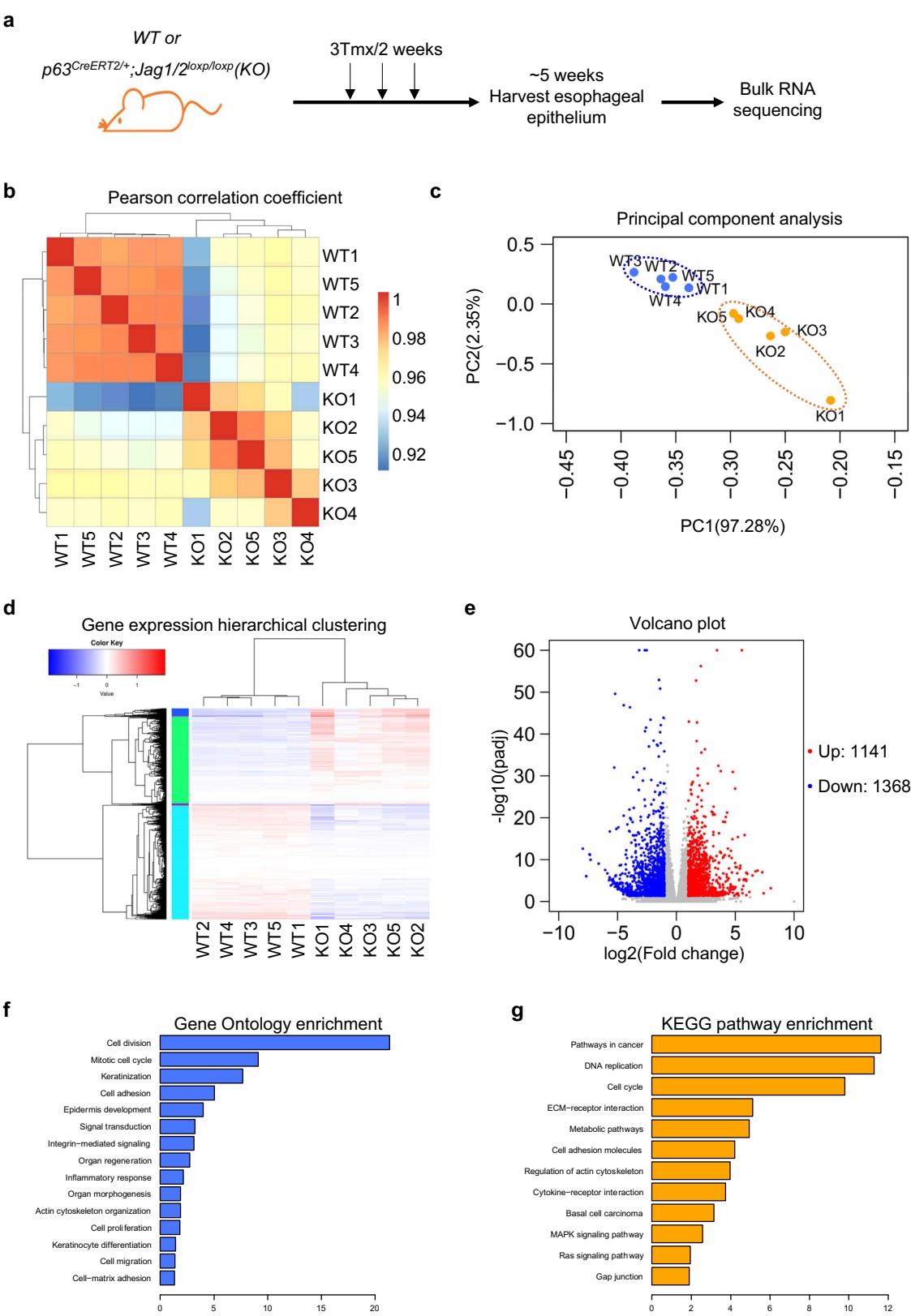

**Fig. 2 | Jag1/2 regulate multiple biological processes and signaling pathways. a** Schematic diagram showing esophageal epithelium harvested from *WT* or *p63^CreERT2/+^;Jag1/2^loxp/loxp^* (*KO*) mice administered with tamoxifen. RNA was isolated from the esophageal epithelium and sequenced. Pearson correlation coefficient analysis (**b**) and principal component analysis (**c**) were performed based on the transcriptional expression profiles of the *KO* versus *WT* samples. **d, e** Differentially expressed genes between *KO* and *WT* mice were presented by hierarchical clustering and a volcano plot. Gene Ontology (**f**) and KEGG pathways (**g**) analysis of the differentially expressed genes between *KO* and *WT* mice. n = 5 per genotype in (**a–g**). *P* values in (**e**, **f**, and **g**) were determined by the two-tailed Wald test, Wallenius' noncentral hypergeometric distribution, and the hypergeometric distribution, respectively. The FDR method was used to calculate an adjusted *P* value ($p_{adj}$).

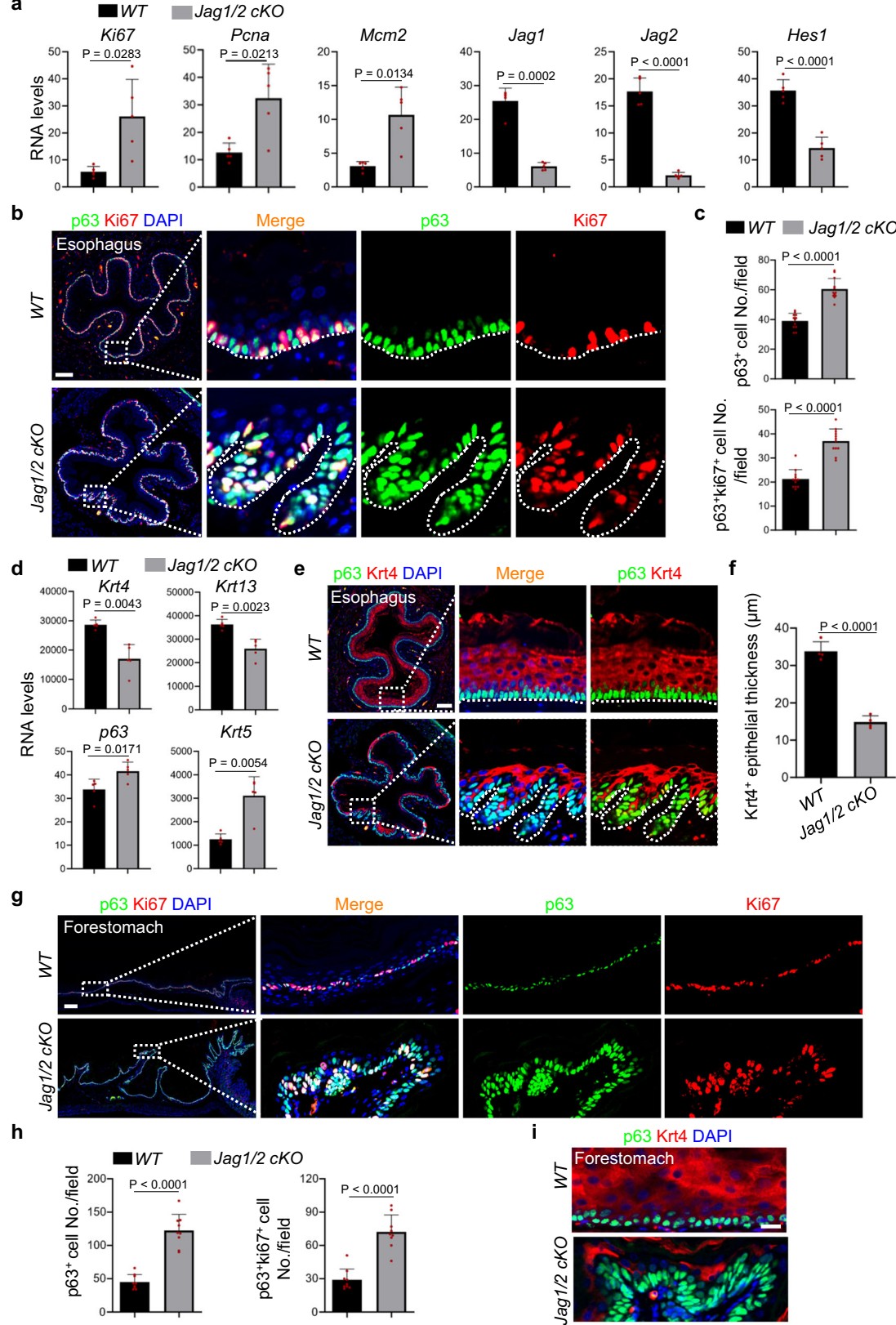

## Jag1/2 represses the initiation of foregut squamous cell carcinoma

Since squamous epithelial dysplasia is a significant preceding lesion of SCC[27], we next determined whether Jag1/2 affect the initiation of SCC. We fed mice drinking water containing 4-Nitroquinoline 1-oxide (4-NQO), a carcinogen that has been widely used to induce SCC in the oral cavity, esophagus, and forestomach in mice[48,49] (Fig. 7a). Strikingly, Jag1/2 KO mice developed severe SCC in the forestomach, with the majority of the mutants dead at around 4-5 weeks (Fig. 7b, c). When tumors developed, Jag1/2 KO mice exhibited illness and stopped food intake, eventually succumbing (Supplementary Fig. 4a). Histological analysis showed that the Jag1/2 KO mice exhibited various features of

**Fig. 3 | The ablation of Jag1/2 dysregulates epithelial homeostasis by perturbing the balance between self-renewal and differentiation of basal progenitor cells. a** Transcript levels of cell proliferation markers *Ki67*, *Pcna* and *Mcm2*, Notch ligands *Jag1/2* and putative Notch downstream target *Hes1* in the mouse esophageal epithelium. Transcript levels were determined by RNA sequencing and presented as FPKM values. Data represent mean ± SD (n = 5 per genotype). Immunofluorescence staining of p63 and Ki67 in the esophageal epithelium (**b**) and quantification of p63+ and p63+Ki67+ cell numbers (**c**). Note the increased p63+ and Ki67+ cells in *Jag1/2 cKO* mice. Data represent mean ± SD (n = 12 per genotype), Scale bar: 100 µm. **d** Transcript levels of suprabasal cell markers *Krt4* and *Krt13* and basal cell markers *p63* and *Krt5* in the esophageal epithelium. Transcript levels were determined by RNA sequencing and presented as FPKM values.

Data represent mean ± SD (n = 5 per genotype). Immunofluorescence staining of p63 and Krt4 in the esophageal epithelium (**e**) and quantification of the Krt4+ esophageal epithelium thickness (**f**). Note the decreased Krt4+ epithelial thickness in the *Jag1/2 cKO* mice. Data represent mean ± SD (n = 4 per genotype), Scale bar: 100 µm. **g, h** Immunofluorescence staining of p63 and Ki67 in the forestomach epithelium (**g**) and quantification of p63+ and p63+Ki67+ cell numbers (**h**). Note the increased p63+ and Ki67+ cell numbers in *Jag1/2 cKO* mice. Data represent mean ± SD (n = 9 per genotype). Scale bar: 200 µm. **i** Immunofluorescence staining of p63 and Krt4 in the forestomach epithelium. Representative images are shown (n = 3 per genotype). Scale bar: 20 µm. *WT wild type*; *Jag1/2 cKO*, p63^CreERT2/+; *Jag1/2^loxp/loxp*. *P* values were calculated by unpaired, two-tailed Student's *t* test.

SCC, including "pearl-like" tissue structures and spindle-shaped tumor cells (Fig. 7d). More than half of the *Jag1/2 KO* mice developed carcinoma in situ or invasive SCC, whereas the majority of the *WT* or the *p63^CreERT2/+* mice only displayed epithelial hyperplasia or dysplasia (Fig. 7e and Supplementary Fig. 4b-d). The tumors highly express multiple SCC diagnostic markers, including p63, Sox2, Krt5, Krt14, Itga6, and Itgb4[50–53] (Supplementary Fig. 5a-e). Moreover, *Jag1/2 KO* mutants also exhibited more severe epithelial dysplasia in the esophagus and tongue (Supplementary Fig. 6a, b). However, no obvious phenotypic changes were observed in the lung and bladder of both the *Jag1/2 KO* mutants and the *control* (Supplementary Fig. 6c, d).

We further characterized the histological features of the forestomach SCC. The proliferation of tumor cells in the forestomach SCC was significantly increased in the mutants (Fig. 7f, g), while epithelial cell differentiation was reduced (Fig. 7h). Furthermore, inflammation is an important feature of tumor initiation[54]. We observed that immune cells were highly elevated in the forestomach SCC of *Jag1/2 KO* mutants (Fig. 8a). These immune cells included F4/80+ or Cd68+ macrophages (Fig. 8b, c), Ly6G+ neutrophils (Fig. 8d), Cd4+ helper T cells (Fig. 8e), and Cd8a+ cytotoxic T cells (Fig. 8f). This is in line with the increased expression levels of various pro-inflammatory cytokines *Il11*, *Il23a*, *Il33*, *Ccl11*, *Ccl22*, and *Tgfβ1* upon Jag1/2 deletion in the mouse esophageal epithelium[55–57] (Supplementary Fig. 7a). The tumor cells were also surrounded by SMA+ stromal fibroblasts, which are also important components of the tumor microenvironment[58] (Supplementary Fig. 7b). Taken together, these results demonstrate that Jag1/2 play an important role in suppressing the initiation of foregut SCC.

**The expression levels of JAG1/2 were lower at the early stage of ESCC carcinogenesis**

Lastly, we investigated the expression of JAG1/2 in the esophageal epithelium at different stages of human ESCC carcinogenesis. We first analyzed the single-cell RNA sequencing data of the esophageal epithelial cells, representing four developmental stages: normal, inflammatory, neoplastic, and ESCC, from available datasets[59–62]. The expression levels of both *JAG1* and *JAG2* were significantly lower at the inflammatory and neoplastic stages, which represent the early stages of ESCC carcinogenesis, compared to the normal state (Fig. 9a, b). However, their expression was higher in the ESCC samples compared to the normal stage (Fig. 9a, b). Consistently, immunostaining showed higher protein levels of both JAG1 and JAG2 in the human ESCC samples (Fig. 9c, d and Supplementary Fig. 8a, b).

## Discussion

In this study, we utilize mouse genetics to demonstrate that Jag1/2 are required to maintain esophageal and forestomach homeostasis while suppressing the initiation of foregut SCC. Conditional deletion of Jag1/2 leads to disruption of the esophageal and forestomach squamous epithelium. Through gene expression profiling, we identify that Jag1/2 regulate various biological processes and signaling pathways. Our subsequent studies reveal that the disruption of epithelial homeostasis is attributed to an increase in proliferation and a decrease in differentiation of basal progenitor cells, which further leads to dilated intercellular space and altered basal cell polarity. Moreover, we reveal that Jag1/2 deficiency impairs Notch signaling activation, leading to reduced epithelial differentiation. Additionally, we demonstrate that *Jag1/2 KO* exacerbates inflammation and epithelial hyperplasia in the forestomach when exposed to bile acids. Notably, our study uncover that Jag1/2 ablation accelerates the initiation of foregut SCC, characterized by enhanced epithelial dysplasia and inflammation. Finally, we show that the expression of JAG1/2 is lowered in the early stages of human ESCC carcinogenesis.

Prior studies showed that the NOTCH1/3 receptors are required for esophageal squamous epithelial differentiation[36]. Our recent investigations have further elucidated the importance of Jag1/2 in promoting squamous stratification during the development of the embryonic esophageal epithelium in mice[37]. However, the role of Notch ligands in esophageal homeostasis remains largely unclear. Here, we employed RNA sequencing and immunostaining to identify that Jag1/2 are most abundantly expressed in both the mouse esophageal and forestomach epithelium, as compared to Dll1/3/4. Genetic ablation of Jag1/2 in the esophageal epithelium and forestomach, the counterpart of the human distal esophagus[12,13], led to abnormal epithelial structures, which were characterized by basal cell hyperplasia and dilated epithelial intercellular space. These results support the essential role of Jag1/2 in maintaining the homeostasis of the esophageal and forestomach squamous epithelium.

We then conducted bulk RNA sequencing and identified more than 2000 genes that exhibited at least a 2-fold change in expression levels. These genes are involved in multiple biological processes, such as cell proliferation, keratinization, cell differentiation, and cell-matrix adhesion, which are critical for maintaining the homeostasis of the esophageal epithelium[2]. Histological analysis further revealed a significant increase in basal progenitor cell proliferation but reduced differentiation in *Jag1/2 KO* mutants. The balance between basal progenitor cell proliferation and differentiation has been established as a requirement for squamous epithelial homeostasis[1–3,63]. Consequently, the disruption of this equilibrium resulting from the Jag1/2 deletion leads to significant deregulation of homeostasis within the squamous epithelium. Moreover, in *Jag1/2 KO* mice, there was a noticeable upregulation in the expression of basal cell adhesion molecules, accompanied by the presence of dilated intercellular space and disrupted basal cell polarities. These observed changes are likely attributed to the perturbed equilibrium between the self-renewal and differentiation processes of basal progenitor cells.

Subsequent GSEA analysis unveiled a reduction in the Notch signaling pathway due to Jag1/2 deficiency. The decrease in the protein levels of NICD1 and the number of nuclear Notch1+ cells in the esophageal epithelium further confirmed impaired Notch signaling activation with Jag1/2 deletion. Interestingly, Jag1/2 proteins were mainly expressed in the basal cells of the esophageal epithelium. Notch1 localized to the membrane of basal cells, but primarily in the nucleus of suprabasal cells. We hypothesize that Jag1/2 on basal cells activate Notch signaling in suprabasal cells to promote their differentiation.

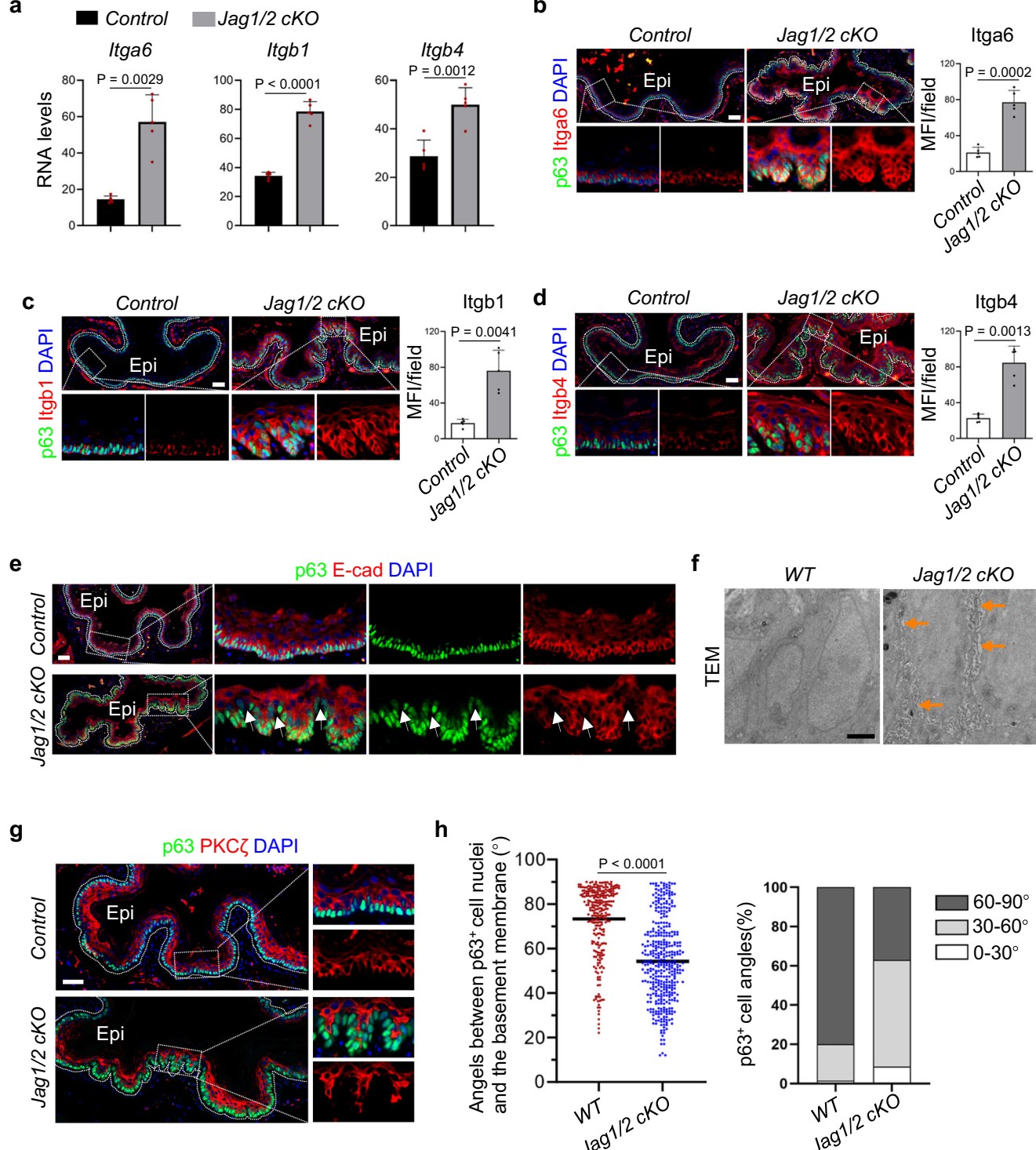

**Fig. 4 | The absence of Jag1/2 results in elevated cell adhesion molecule expression, dilated intercellular space, and aberrant basal progenitor cell polarity. a** Transcript levels of basal cell adhesion molecules *Itga6*, *Itgb1* and *Itgb4* determined by RNA sequencing and presented as FPKM values. Data represent mean ± SD (n = 5 per genotype). **b-d** Immunofluorescence staining of Itga6, Itgb1, Itgb4, and p63. Cells above the white dotted lines are epithelium. Note the increased expression levels of Itga6, Itgb1 and Itgb4 in *Jag1/2 cKO* mice. Data represent mean ± SD (n = 5 per genotype). Epi, epithelium. MFI mean fluorescence intensity. Immunostaining of E-cadherin (E-cad) and p63 (**e**) and transmission electron microscopy (TEM) of the esophageal epithelium (**f**). Representative images are shown (n = 4 in (**e**) per genotype, n = 3 in (**f**) per genotype). Note the increased intercellular space between basal cells (white and orange arrows) in *Jag1/2 cKO* mice. Epi, epithelium. Scale bars: **e** 50 μm; **f** 2 μm. **g** Immunostaining of PKCζ and p63 in the esophageal epithelium in *Jag1/2 cKO* mice. Epi epithelium. Note the disorganized expression pattern of PKCζ. Representative images are shown (n = 4 per genotype). Scale bar: 50 μm. **h** Angles between the long axes of the p63⁺ basal cell nuclei and the basement membrane (n = 266 for *WT*, n = 390 for *Jag1/2 cKO*). *P* values were calculated by unpaired, two-tailed Student's *t* test. *Control, WT* or *p63^{CreERT2/+}; Jag1/2 cKO, p63^{CreERT2/+};Jag1/2^{loxp/loxp}*.

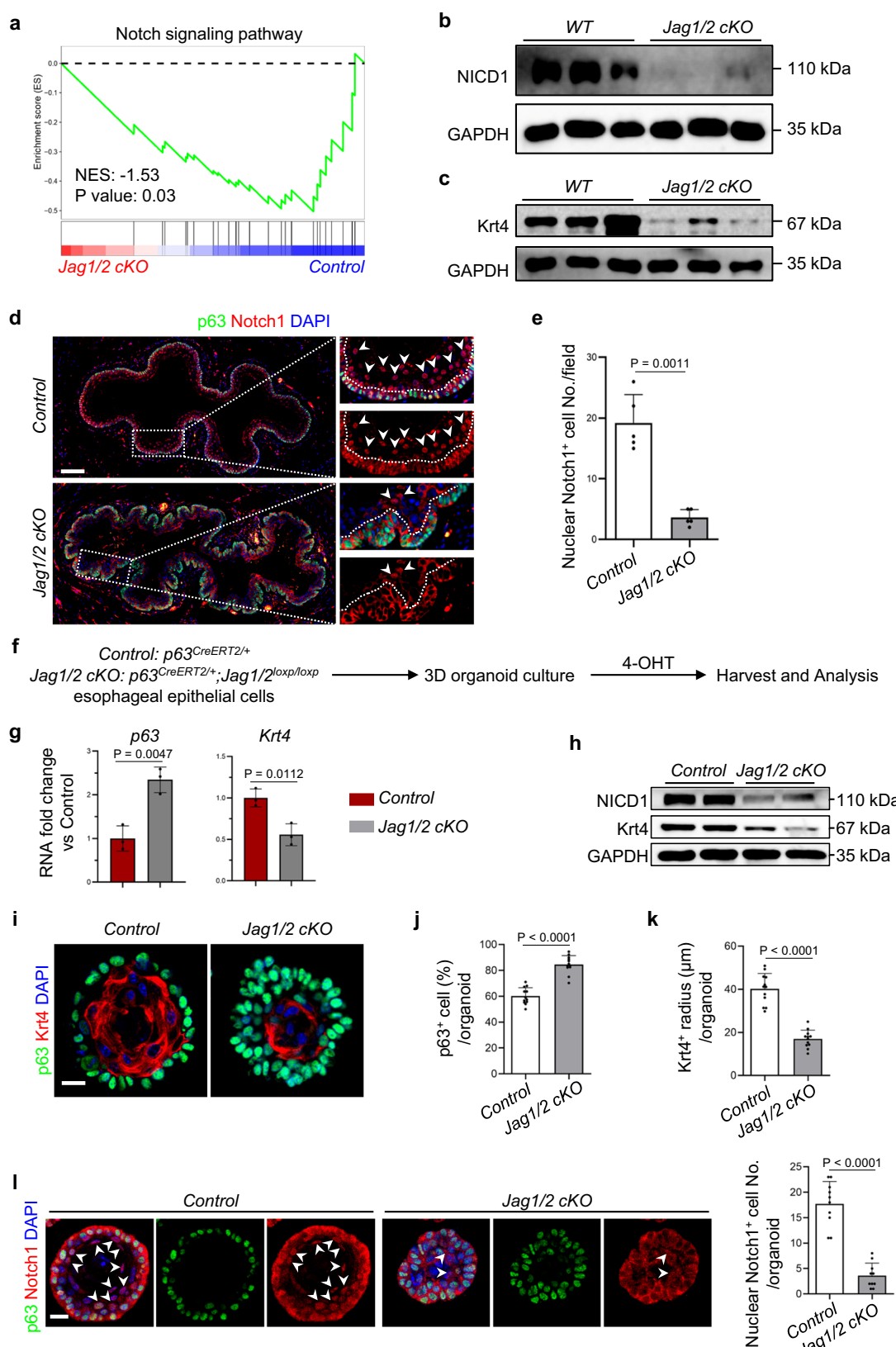

Previous studies have shown that Notch signaling negatively regulates the expression of the basal cell master transcription factor p63, thereby promoting cell differentiation in skin keratinocytes[64]. Additionally, p63 has been shown to directly transactivate the expression of integrins, including Itga6, Itgb1, and Itgb4[65]. Hence, the decrease in differentiation and upregulation of the integrins in the foregut squamous epithelium is likely attributed to the elevated expression levels of p63 upon Jag1/2 deletion. Furthermore, we established 3D organoids from mouse esophageal epithelial cells and showed that Notch signaling and epithelial cell differentiation are downregulated with Jag1/2 deficiency in the organoids, indicating a direct regulatory role of Jag1/ 2 in the cell differentiation within the esophageal the epithelium.

**Fig. 5 | The deficiency of Jag1/2 reduces squamous epithelial differentiation through impaired Notch signaling. a** Gene set enrichment analysis (GSEA) showed that Notch signaling was downregulated in *Jag1/2 cKO* mutants. Western blot showed that NICD1 (**b**) and the differentiation marker Krt4 (**c**) were downregulated in the esophageal epithelium of *Jag1/2 cKO* mutants (n = 3 per genotype). Immunostaining of Notch1 and p63 (**d**) and quantification of nuclear Notch1[+] cells (**e**). Note that the number of nuclear Notch1[+] cells was reduced in the *Jag1/2 cKO* mutants and that Notch1 mainly localizes in the nuclei of suprabasal cells (arrows), in contrast to the enrichment in the plasma membrane of basal cells. Cells above the white dotted lines are suprabasal cells. Data represent mean ± SD (n = 5 per genotype). Scale bar: 100 μm. *Control, WT* or *p63^{CreERT2/+}*. **f** Schematic diagram showing the generation of 3D organoids from the esophageal epithelial cells of *control p63^{CreERT2/+}* or *p63^{CreERT2/+};Jag1/2^{loxp/loxp}* (c*KO*) mice. Gene deletion was induced by adding 4-hydroxytamoxifen (4-OHT). **g** RNA expression fold change of *p63* and *Krt4* determined by qPCR. Data represent mean ± SD (n = 3 per genotype). **h** Western blot showed a reduction in the expression levels of NICD1 and Krt4 in *Jag1/2 cKO* organoids. Shown are representative data (n = 4 per genotype). **i–k** Immunostaining analysis showed an increase in p63-expressing basal cells and a decrease in Krt4[+] squamous differentiation in the *Jag1/2 cKO* organoids. Scale bar: 20 μm. Data represent mean ± SD (n = 12 per genotype). **l** Immunostaining analysis showed a reduction in the number of nuclear Notch1[+] cells in *Jag1/2 cKO* organoids. Data represent mean ± SD (n = 10 per genotype). Scale bar: 20 μm. *P* values were calculated by unpaired, two-tailed Student's *t* test. *Jag1/2 cKO, p63^{CreERT2/+};Jag1/2^{loxp/loxp}*.

Epithelial integrity is critical for protecting the squamous epithelium from injury induced by various insults[45]. To investigate the effect of Jag1/2 deficiency on injury development, mice were treated with DCA water. Our results showed that the absence of Jag1/2 significantly exacerbated squamous epithelial injury in the forestomach, resulting in more severe inflammation and epithelial dysplasia. Given that compromised epithelial integrity allows easier entry of pathogens such as bacteria[66], it will be significant to determine whether the foregut of *Jag1/2 KO* mice is more susceptible to pathogenic infections in future studies. Notably, *Jag1/2 KO* mutants exhibited basal cell hyperplasia, a critical characteristic observed in the ESCC[24]. To further determine whether Jag1/2 modulate foregut SCC initiation, we utilized a 4-NQO-induced foregut SCC mouse model. Our results demonstrated that *Jag1/2 KO* led to the early onset of foregut SCC, characterized by severe epithelial dysplasia and highly proliferative cancer cells. Considering that inflammation is a well-known hallmark of SCC[22], it is noteworthy that we also observed substantial inflammation in the SCC of *Jag1/2 KO* mutants, with the presence of various types of immune cells, such as macrophages, neutrophils, and T cells. These findings are consistent with the increased expression of multiple pro-inflammatory cytokines upon *Jag1/2* deletion. Consistently, we observed lower expression levels of both JAG1 and JAG2 in the early stages, including inflammatory and neoplastic states, of human ESCC carcinogenesis. Our findings corroborate previous studies that have established Notch signaling as a tumor suppressor of squamous cell carcinoma (SCC) initiation[30,38–40]. Nevertheless, contrasting evidence suggests a pro-tumorigenic role of Notch signaling during SCC progression[41,42]. In line with this, we found that the expression levels of JAG1/2 were elevated in the human ESCC samples compared to normal samples. It is possible that Jag1/2-mediated Notch signaling exerts distinct functions at different stages of ESCC carcinogenesis, which is suppressing the onset of ESCC while promoting its progression. However, validating these hypotheses necessitates further investigation through comprehensive mouse genetic studies by modulating Jag1/2-Notch signaling at various stages of carcinogenesis. Taken together, our studies demonstrate that Jag1/2 protect the foregut from injury and SCC carcinogenesis.

Overall, our study has identified an essential role for Jag1/2-mediated Notch signaling in maintaining the homeostasis of the esophageal and forestomach epithelium by regulating the balance between self-renewal and differentiation of basal progenitor cells. Moreover, we have demonstrated that Jag1/2 protect mice from foregut injury and suppress the onset of SCC in the foregut. Our studies provide valuable insights into the mechanisms that link the function of basal progenitor cells to the maintenance of esophageal and foregut homeostasis and the initiation of SCC.

## Methods

This study complies with all relevant ethical regulations approved by the Institutional Review Board of Shanghai Jiao Tong University and the ethics committee of Fujian Medical University Union Hospital.

### Mice

*p63^{CreERT2/+}, Jag1^{loxp/loxp} and Jag2^{loxp/loxp}* mouse lines have been previously described[67–69]. Mice were maintained on a mixed background of C57BL/6 and 129SvEv, and both sexes were used. Mice aged between 6 and 12 weeks were utilized in the studies. The mice were kept in specific-pathogen-free conditions with a 12-h light/night cycle and an ambient temperature of 20–24 °C with 40–60% humidity. Mice were provided with food and water ad libitum. Tail DNA was used for mouse genotyping. All experimental procedures were conducted in accordance with protocols approved by the Institutional Animal Care and Use Committee (IACUC) of Shanghai Jiao Tong University. All experimental mice were euthanized when their body weight loss exceeded 20%. The tumor volume of all mice did not exceed the maximum size of 2000 mm³ permitted by the IACUC of Shanghai Jiao Tong University.

### DCA-induced mouse forestomach epithelial injury

*p63^{CreERT2/+};Jag1/2^{loxp/loxp}* and control mice were administered tamoxifen at a dose of 100 mg/kg body weight/day for three consecutive days via intraperitoneal injection. Two days after the last dose of tamoxifen injection, mice were fed DCA (Sigma-Aldrich, 30970) water (3%, pH = 7.0) via oral gavage at a dose of 100 mg/kg body weight/day every other day for 3 days. Tissues were harvested three days after the last feeding. Histological features of dysplasia were scored based on the degree of changes in the tissues by previously established criteria: 0, normal; 1, mild dysplasia; 2, moderate dysplasia; 3, severe dysplasia[10,70].

### Mouse model of foregut SCC

To induce SCC in the forestomach, *p63^{CreERT2/+};Jag1/2^{loxp/loxp}* and control mice were injected with tamoxifen at a dose of 100 mg/kg body weight/day for three consecutive days. Two days after the last dose of tamoxifen administration, mice were fed with 100 μg/ml 4-NQO (Sigma-Aldrich, N8141) drinking water.

### Human tissue sections

The study on human ESCC tissue arrays was approved by the ethics committee of Fujian Medical University Union Hospital. Tissue arrays comprising 44 human ESCC samples and 42 adjacent normal esophageal epithelial tissues from 50 to 78-year-old male and female ESCC patients were collected at Fujian Medical University Union Hospital, and subsequent immunohistochemistry (IHC) was conducted on the tissue sections. All samples were obtained with informed consent from all subjects and in accordance with the ethical standards of the same institute. The specimen diagnosis was conducted by pathologists. IHC scores were determined on a scale of 0–3 (0, negative; 1 low; 2, moderate; 3, high).

### 3D organoid culture

The esophagus from *p63^{CreERT2/+};Jag1/2^{loxp/loxp}* or *p63^{CreERT2/+}* control mice was dissected, and the muscle layers were peeled off using forceps. Following this, the esophagus was incubated with 0.5 U/ml Dispase I solution for 10 minutes, and the epithelium was separated from the

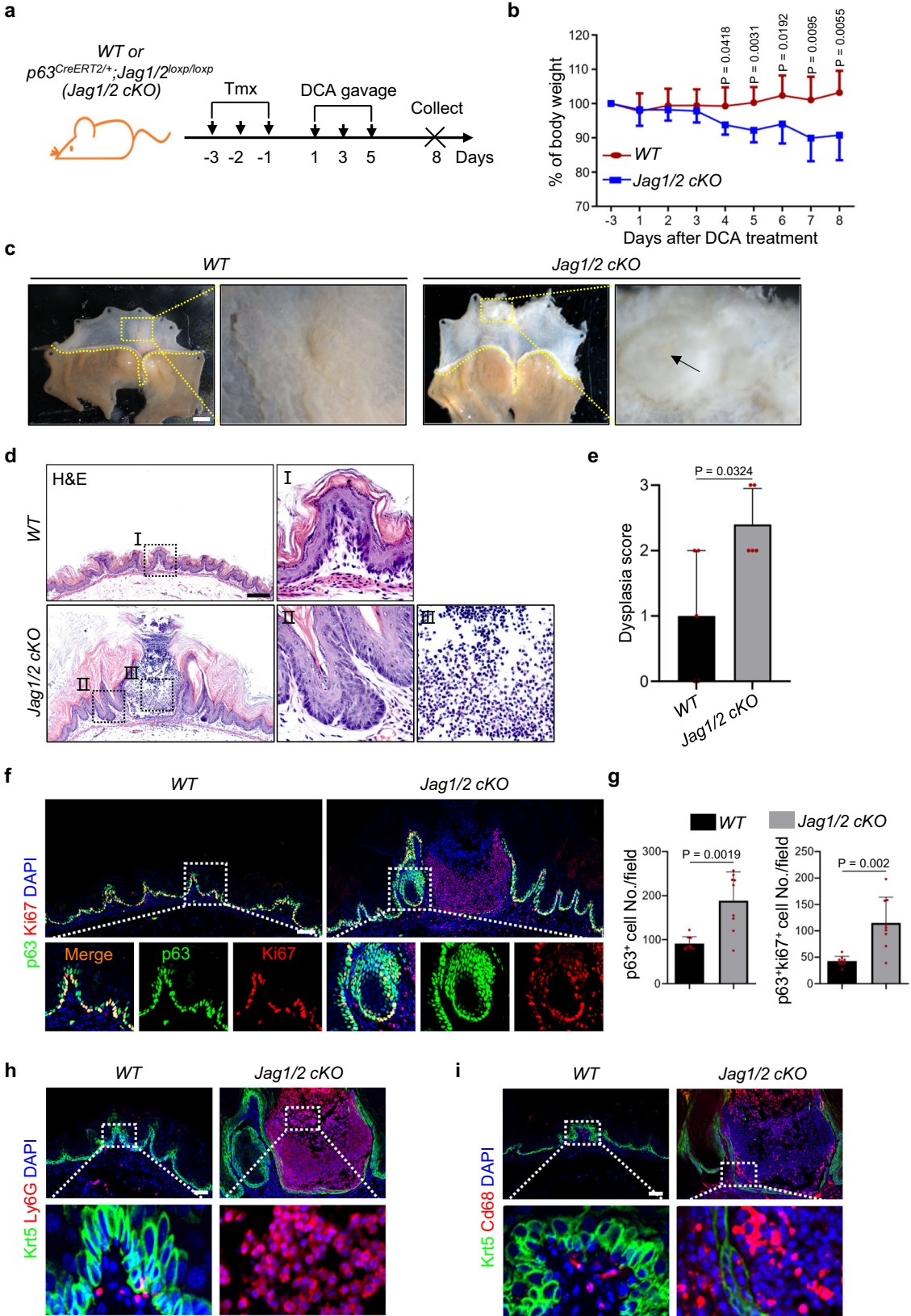

mesenchyme using forceps. Subsequently, the epithelium was cut into small pieces and further digested into single cells with 1 mg/ml Collagenase I and Collagenase II. The single cells were then embedded in Matrigel and cultured in organoid medium: Advanced DMEM/F12 (Thermo Fisher Scientific, 12634010) containing 10 mM HEPES (Thermo Fisher Scientific, 15630080), 1X B27 (Thermo Fisher

Scientific, 17504044) supplement, 1X GlutaMAX (Thermo Fisher Scientific, 35050061), 1 mM N-Acetylcysteine (Selleck Chemicals, S1623), 5 mM Nicotinamide (Selleck Chemicals, S1899), 3 μM CHIR99021 (Tocris, 4423), 100 ng/ml EGF (R&D System, 236-EG), 100 ng/ml FGF10 (R&D System, 345-FG), 0.5 μM A83-01 (Selleck Chemicals, S7692), 1 μM SB202190 (Selleck Chemicals, S1077), 0.1 μM SAG (Selleck Chemicals,

**Fig. 6 | Jag1/2 deficiency exacerbates squamous epithelial injury in the forestomach. a** Schematic illustrating deoxycholic acid (DCA)-induced inflammatory diseases in the forestomach. *WT, wild type; Jag1/2 cKO, p63^CreERT2/+;Jag1/2^loxp/loxp*. **b** Body weight change of the mice gavaged with DCA water. Note the significantly reduced body weight of the *Jag1/2 cKO* mutants as compared to the *WT* control mice. Data represent mean ± SD (n = 7 per genotype). **c** Representative gross morphology of the forestomach. Note the mucosal hyperplasia in the forestomach of *Jag1/2 cKO* mice. Scale bar: 2 mm. **d** Representative H&E stained forestomach sections of *WT* and *Jag1/2 cKO* mice gavaged with DCA water. Scale bar: 200 μm. **e** Histological scores of dysplasia in the forestomach of mice gavaged with DCA water. Data represent mean ± SD (n = 5 per genotype). **f** Immunofluorescence staining of p63 and Ki67 in the forestomach of mice gavaged with DCA water. Scale bar: 100 μm. **g** Quantification of p63+ and p63+Ki67+ cell numbers. Note the increased p63+ and Ki67+ cells in the *Jag1/2 KO* mice. Data represent mean ± SD (n = 9 per genotype). Immunofluorescence staining of Krt5, Ly6G+ neutrophils, and Cd68+ macrophages in the forestomach of mice gavaged with DCA water. Note that neutrophils and macrophages were increased in *Jag1/2 cKO* mice gavaged with DCA water. Representative images are shown (n = 4 in (**h**) per genotype, n = 3 in (**i**) per genotype). Scale bars: 50 μm. *P* values were calculated by unpaired, two-tailed Student's t-test.

S7779), 2 μM Dorsomorphin dihydrochloride (Selleck Chemicals, S7306), 5 μM SB431542 (Tocris, 1614), 10 μM Y-27632 (Tocris, 1254), 1X Penicillin-Streptomycin (Thermo Fisher Scientific, 15140122), 50 μg/ml Gentamicin (Thermo Fisher Scientific, 15710064), and 250 ng/ml Amphotericin B (Sigma-Aldrich, A2942). Two days after seeding, 0.5 μM 4-OHT (Sigma-Aldrich, H-6278) was added to the organoids to induce Jag1/2 deletion. The organoids were harvested for analysis 7 days post-tamoxifen treatment.

### H&E staining
Paraffin-embedded slides were first deparaffinized with xylene, followed by a series of grades of ethanol, and finally with water. The tissues on the slides were then incubated with a hematoxylin solution for one minute and washed with tap water until the water was clear. Next, tissues were counterstained in an eosin solution for 15 s. After counterstaining, tissue slides were immediately transferred to 95% ethanol and further dehydrated with 100% ethanol and xylene. Finally, slides were mounted with a neutral balsam mounting medium and air-dried before imaging.

### Evans blue staining to assess mucosal permeability
A solution of Evans blue (Sigma-Aldrich, E2129) (1% in 1X PBS) was injected into the esophagus and stomach, with the distal end of the stomach securely tied off. Subsequently, the proximal end of the esophagus was also tied off. The tissues were then immersed in 1X PBS for four hours. Afterwards, both tied ends were opened, and the tissues were fixed in paraformaldehyde before being sectioned for histological analysis.

### Immunostaining
For immunofluorescence staining, paraffin slides were first deparaffinized with xylene, then hydrated in a series of graded ethanol, and finally with water. Antigen retrieval was then performed in a Tris-HCl buffer (pH = 9.0) using a pressure cooker. The tissues were blocked with a blocking buffer (1X PBS containing 3% donkey serum and 0.3% Triton X-100) for one hour. Primary antibodies, listed in Supplementary Table 1, were diluted in the blocking buffer at a ratio of 1:200. The diluted primary antibodies were added to the tissues, which were then stored in a 4 °C refrigerator overnight. The next day, tissues were washed with 1X PBS for 30 min. Secondary antibodies conjugated to Alexa Fluor 488, Cy3, or Cy5 (Jackson ImmunoResearch Laboratories) were diluted in 1X PBS and added to the slides. ImageJ (National Institutes of Health) (v1.53c) was used for the quantitative analysis of the immunofluorescence-stained images. For IHC staining, following primary incubation, tissue sections were incubated with secondary antibodies conjugated to HRP and then reacted with DAB substrates. Secondary antibodies are listed in Supplementary Table 2.

### Microscopy imaging
Images of immunofluorescence-stained tissue slides were acquired by a Nikon Ni-E A1 HD25 confocal microscope, and images of H&E-stained slides were obtained using a ZEISS Axio Imager 2 microscope. For imaging the gross morphology of large tissues such as the forestomach, Olympus MVX10 stereoscopic microscope was utilized. Transmission electron microscopy images were acquired using a Thermo Fisher Scientific Talos L120C G2 transmission electron microscope.

### Isolation of the esophageal epithelium
The esophagus was dissected from the mice and muscle layers and adventitia were stripped away using forceps. Subsequently, the esophagus was incubated with 0.5 U/ml of Dispase I (Sigma-Aldrich, D4818) at room temperature for 10 min, allowing the separation of mesenchymal layers from the epithelium. The mesenchyme layers were then removed from the epithelium under a dissection microscope, and the epithelium was then washed with sterile 1X PBS and lysed with RNA lysis buffer. Total RNA was extracted from the lysate using the Qiagen RNeasy Plus Universal Mini Kit.

### RNA sequencing
Libraries were constructed using the VAHTS Universal V8 RNA-seq Library Prep Kit for Illumina (Vazyme Biotech) and size-selected with VAHTS DNA Clean Beads (Vazyme Biotech) according to the manufacturer's instructions. The libraries were sequenced by Illumina NovaSeq 6000 using a paired-end (150 bp for each end of the fragment) configuration. The raw sequencing data, including Read1 and Read2 FASTQ files, were processed by Cutadapt (v1.9.1, Phred score cutoff: 20, Maximum error rate: 0.1, Minimum overlap: 1 bp, minimum sequence length: 75, proportion of N: 0.1) to obtain high-quality clean reads. The reads were then mapped to a reference genome (Mouse: UCSC/mm9) using HISAT2 (v2.0.1). Gene expression levels were estimated by HTSeq (v0.6.1) and presented in read counts or Fragments Per Kilobase of transcript per Million mapped reads (FPKM). Pearson correlation coefficient analysis and principal component analysis (PCA) were performed on the gene expression data presented in FPKM for all samples.

### Differential expression analysis
The gene read counts were analyzed by the DESeq2 Bioconductor package (v1.26.0) to determine differentially expressed genes (DEGs) between WT and mutant samples. The fold change and p-values for each gene were calculated, and a false discovery rate (FDR) error method was employed to perform multiple hypothesis test correction. Genes with expression fold change ≥2 and FDR ≤ 0.05 were determined as DEGs. DEGs were then visualized using a volcano plot and a hierarchical clustering heatmap.

### Gene ontology enrichment and KEGG pathway enrichment analyses
The GOseq Bioconductor package (v1.34.1) was utilized to identify Gene Ontology (GO) terms in biological processes that annotate a list of enriched genes with FDR ≤ 0.05. Enrichment analysis of the DEGs in KEGG pathways[71] was performed by the ClusterProfiler Bioconductor package[72]. Pathways with FDR ≤ 0.05 were considered significantly enriched.

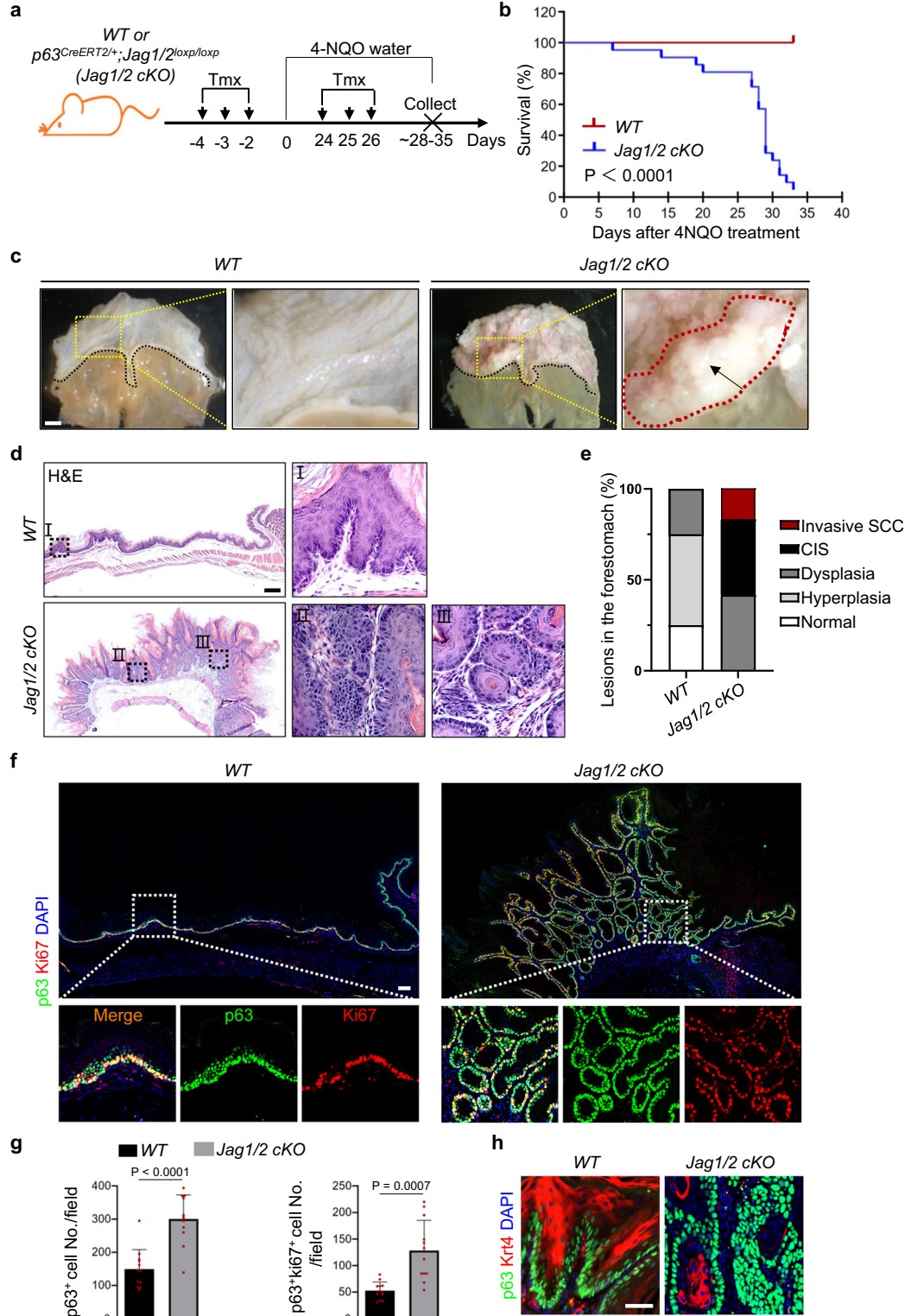

**Fig. 7 | Jag1/2 represses the initiation of foregut squamous cell carcinoma.**
**a** Schematic illustrating 4-NQO-induced squamous cell carcinoma in the forestomach. *WT*, *wild type*; *Jag1/2 cKO*, *p63 p63^CreERT2/+;Jag1/2^loxp/loxp*. **b** Survival curves of *WT* and *Jag1/2 cKO* mice fed with 4-NQO water. Note the lower survival rate of *Jag1/2 cKO* mice compared with *WT* mice (n = 21 per genotype). *P* value was determined by the log-rank (Mantel-Cox) test. **c** Representative gross morphology of the forestomach of mice fed with 4-NQO water. Note the tumors formed in the forestomach of *Jag1/2 cKO* mice. Scale bar: 2 mm. **d** Representative H&E-stained forestomach sections of *WT* and *Jag1/2 cKO* mice fed with 4-NQO water. Note the carcinoma in situ that formed in the *Jag1/2 cKO* mice. Representative images are

shown (n = 12 per genotype). Scale bar: 200 μm. **e** Percentage of the indicated types of lesions in the forestomach of mice fed with 4-NQO water (n = 12 per genotype). CIS, carcinoma in situ; SCC, squamous cell carcinoma. **f** Immunofluorescence staining of p63 and Ki67 on the forestomach of mice fed with 4-NQO water. Note the highly increased p63+ and Ki67+ cells in the *Jag1/2 KO* mice. Scale bar: 100 μm. **g** Quantification of p63+ and p63+Ki67+ cells. Data represent mean ± SD (n = 12 per genotype). *P* values were calculated by unpaired, two-tailed Student's *t* test. **h** Immunofluorescence staining of p63 and Krt4 in the forestomach of mice fed with 4-NQO. Representative images are shown (n = 3 per genotype). Scale bar: 50 μm.

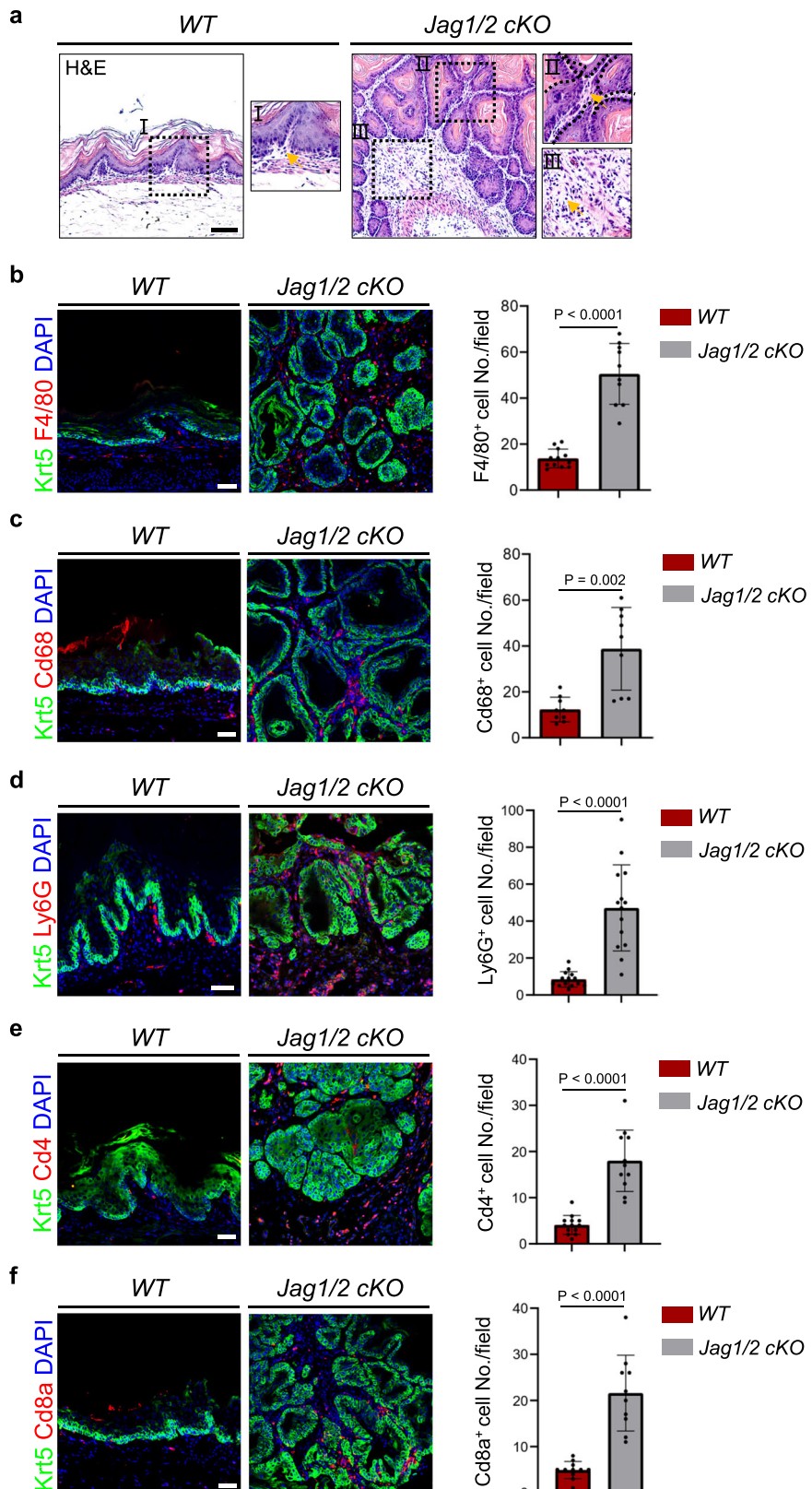

**Fig. 8 | The ablation of Jag1/2 results in an increase in the accumulation of immune cells in foregut squamous cell carcinoma. a** H&E-stained sections show increased immune cells in the forestomach SCC in *Jag1/2 cKO* mice fed 4-NQO water compared to WT mice. Representative images are shown (n = 12 per genotype). Scale bar: 100 μm. Immunofluorescence staining of Krt5, F4/80+ (n = 12 for *WT*, n = 10 for *Jag1/2 cKO*) (**b**) and Cd68+ macrophages (n = 9 per genotype) (**c**), Ly6G+ neutrophils (n = 14 per genotype) (**d**), Cd4+ helper T cells (n = 13 for *WT*, n = 11 for *Jag1/2 cKO*) (**e**), and Cd8a+ cytotoxic T cells (n = 11 for *WT*, n = 10 for *Jag1/2 cKO*) (**f**). Data represent mean ± SD. *P* values were calculated by unpaired, two-tailed Student's t-test. Scale bars: 50 μm. *WT, wild type; Jag1/2 cKO, p63^CreERT2/+;Jag1/2^loxp/loxp*.

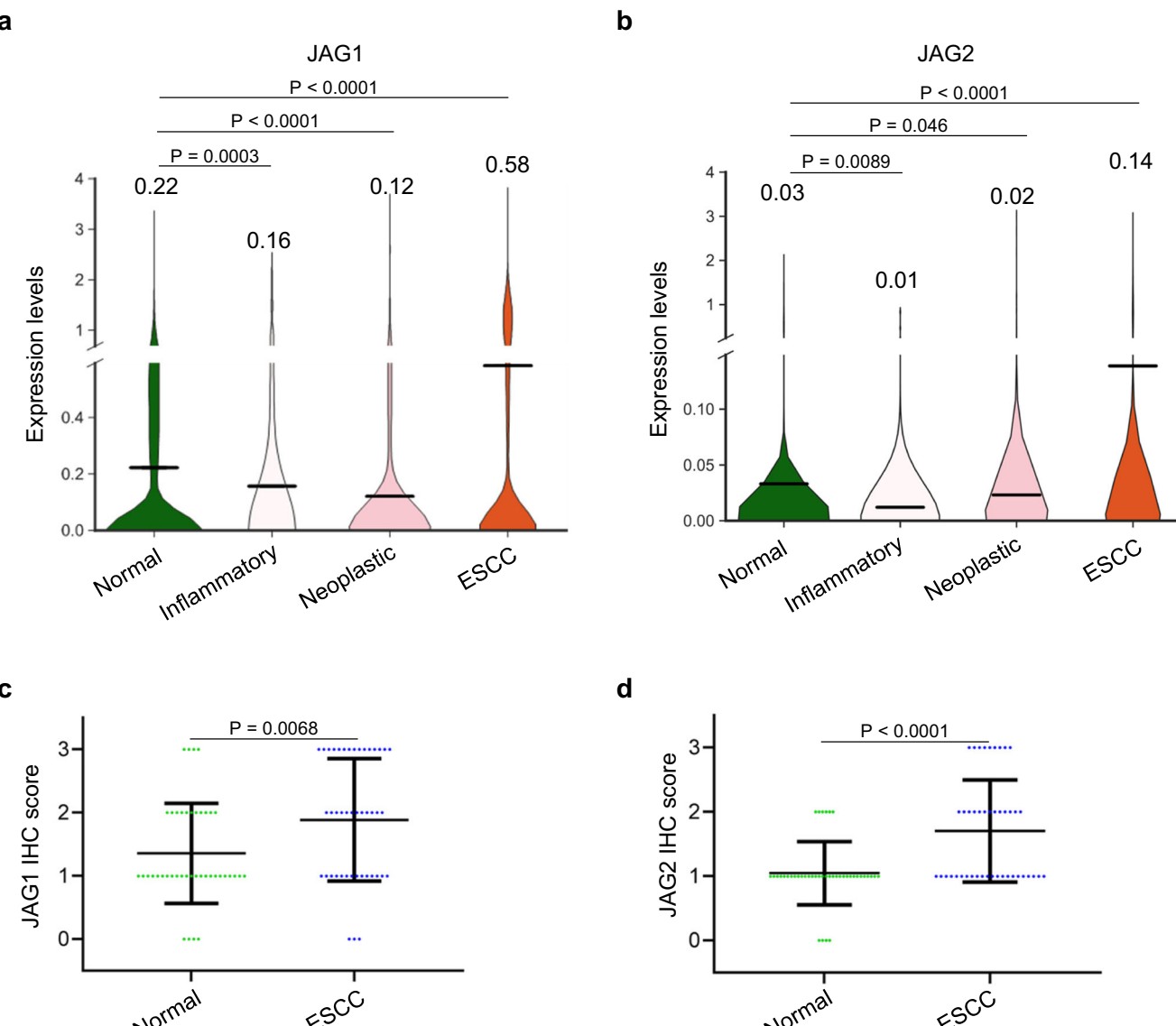

**Fig. 9 | The expression of JAG1 and JAG2 is downregulated during tumor initiation stages.** Single-cell RNA sequencing of JAG1 (**a**) and JAG2 (**b**) of the esophageal epithelial cells of normal, inflammation, neoplasia, and ESCC (normal, n = 9597; inflammation = 126; neoplasia, n = 1341; ESCC, n = 55,234). Note that the expression levels of JAG1 and JAG2 are both downregulated during inflammation and neoplasia while upregulated in the ESCC samples. IHC scores of JAG1 (**c**) and JAG2 (**d**) in normal and ESCC samples. Data represent mean ± SD (Normal, n = 42; ESCC, n = 44). *P* values were calculated by unpaired, two-tailed Student's *t* test. ESCC esophageal squamous cell carcinoma, IHC immunohistochemistry.

## Single-cell RNA sequencing analysis

The single-cell RNA sequencing data that include four developmental stages of esophageal carcinogenesis: normal, inflammatory, neoplastic, and ESCC, were downloaded from the Gene Expression Omnibus (GEO) (GSE199654[61], GSE160269[59], GSE188955[60], and GSE201153[62]). The data were analyzed using the Python Scanpy package (version 1.9.3). Cells were filtered based on the criteria of expressing 200-8000 genes with a mitochondrial fraction less than 20% and genes detectable in more than three cells. Subsequently, the four datasets were integrated, and batch correction was performed using Scanpy's external harmony. Esophageal epithelial cells were then selected for further analysis.

## Western blot

The proteins were denatured in SDS buffer and separated on a 10–12% SDS-polyacrylamide gel. Subsequently, they were transferred to a PVDF membrane and incubated with a primary antibody in a 5% milk TBST-blocking buffer overnight. The following day, the membranes were washed and incubated with a secondary antibody conjugated to HRP for 2 h. After incubation, the membranes were washed and developed using SuperSignal™ West Pico PLUS Chemiluminescent Substrate. Images were acquired with a ChemiDoc Imaging system (Bio-Rad). Primary and secondary antibodies are listed in Supplementary Table 1 and Supplementary Table 2, respectively.

## Quantitative PCR (qPCR)

Total RNA was extracted using the RNeasy Plus Universal Mini Kit (Qiagen, 73404) and reverse transcribed using the SuperScript IV First-Strand Synthesis System (Thermo Fisher Scientific, 18091050). Amplifications were performed in the qTOWER³ real-time PCR Thermal Cyclers (Analytik Jena) using SYBR Green Supermix (Bio-Rad, 1725120). Transcript levels of all genes were normalized to β-actin using the $2^{(-\Delta\Delta CT)}$ method. Fold changes for all genes were presented as

the fold change of the indicated samples. qPCR primer sequences are listed in Supplementary Table 3.

## Quantification and statistical analysis
The data are presented as mean ± SD, and statistical analyses were performed using at least three biological replicates. Unpaired Two-tailed Student's *t* test was calculated by GraphPad Prism 9 (v9.0.0.121) to determine the statistical significance for two groups. A *P* value < 0.05 was considered statistically significant. For Kaplan-Meier survival analysis, *P* values were calculated using the log-rank (Mantel−Cox) test.

## Reporting summary
Further information on research design is available in the Nature Portfolio Reporting Summary linked to this article.

## Data availability
All data from this study are available in the published article and Supplementary Information. Source data are provided with this paper. The RNA-seq data generated in this study are publicly available in the Genome Sequence Archive in National Genomics Data Center, China National Center for Bioinformation / Beijing Institute of Genomics, Chinese Academy of Sciences database under accession code CRA011089 (https://ngdc.cncb.ac.cn/gsa). The UCSC/mm9 mouse reference genome used in this study is available at http://genome.ucsc.edu/cgi-bin/hgGateway?db=mm9. Source data are provided with this paper.

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

## Acknowledgements

We thank all members of the Zhang lab for their critical reading of the manuscript. We also thank the Core Facility and Technical Service Center, School of Life Science and Biotechnology at Shanghai Jiao Tong University. This work was supported by grants from the National Key Research and Development Program of China (2022YFA0912600 to L.Z., C.Y., B. Chu and Y.Z.), the National Natural Science Foundation of China (32170831 to Y.Z.), the Natural Science Foundation of Shanghai (22ZR1435300 to Y.Z.), the National Key Research and Development Program of China (2021YFA1100400 to Q.W.), and the National Natural Science Foundation of China (32070865 to Q.W.).

## Author contributions

Conceptualization, H.H., Y.J., Q.W., and Y.Z.; methodology, investigation, and validation, H.H., Y.J., J. Liu, D.L., J.Y., R.M., X.L., Q.W., and Y.Z.; software and bioinformatic analysis, D.L., X.Y., Y.Y.; writing, review and editing, H.H., Y.J., Q.W., and Y.Z.; reagents, D.S., J. Lin, Q.C., M.J., J.X., research advice and discussion, H.H., Y.J., B. Chu, C.Y., L.Z., B. Cao, Q.W., Y.Z.; supervision, Q.W., and Y.Z.; funding acquisition L.Z., C.Y., B. Chu, Q.W. and Y.Z.

## Competing interests

The authors declare no competing interests.
