## [Peer Review File · Nature Communications]

Jag1/2 maintain esophageal homeostasis and suppress foregut tumorigenesis by restricting the basal progenitor cell poolREVIEWER COMMENTS

Reviewer #1 (Remarks to the Author):

This manuscript by Huang et al carefully characterized the morphological and molecular phenotypes in Jag1/2KO mouse esophagus and showed certain relevance to human esophagitis and esophageal squamous cell carcinoma. This study is an extension of their published work on the role of Notch pathway in embryonic esophageal development (Ref#37). It is, in fact, the first one to systematically understand the role of Notch ligands (Jag1/2) in the homeostasis of adult esophageal epithelium and esophageal diseases. This reviewer has several major concerns and minor concerns as follows:

Major concerns:

1. The positional relationship between the ligands, receptors, and effectors of the Notch pathway is not clear. For example, immunofluorescent staining of Jag1, Notch1 (maybe NICD1), and Hes1 may address this issue.
2. Clinical relevance of the findings needs to be clearly defined. For example, how are Jag1/2 and Dll1/3/4 expressed in human esophageal epithelium? Do they follow similar expression patterns as in the mouse esophageal epithelium? Are there any differences in Jag1/2 expression in normal vs GERD esophageal epithelium? Does Jag1/2 downregulation take place in human esophageal squamous cell carcinoma in comparison with the normal esophageal epithelium? Is Jag1/2 downregulation an early or late event during esophageal carcinogenesis? One critical and important issue is whether the mouse models (Figure 5a/6a) recapitulate human diseases or just a biological phenomenon. "Deregulation of homeostasis", "perturbed epithelium", etc are biological terms, not terms in clinical pathology. A clinical pathologist may be consulted regarding these issues related to clinical relevance.
3. Lack of mechanistic in-depth is a concern. For example, how does Jag1/2KO impact the PI3K pathway? Why does Jag1/2KO cause body weight loss and death when the mice were given 4NQO (Figure 5b/6b)? Why does Jag1/2KO cause inflammation? If compromised epithelial barrier function and subsequent bacterial infection are important factors, additional experimental data need to be presented, for example, using germ-free mice.
4. Dilated intercellular space and leaky epithelium are possible consequences of Jag1/2KO. TEM and TEER analysis are needed to validate these structural and functional phenotypes in the esophageal epithelium.

Minor concerns:

1. Protein names and gene names need to be expressed consistently and follow the requirements of the journal. For example, most cancer journals may require protein names to be capitalized.
2. DCA should be deoxycholic acid instead of dichloroacetate. DCA is rarely detected in the upper gastrointestinal tract in humans. Treatment with gastric content (for example, acid, enzymes, conjugated bile acids) is more clinically relevant than DCA treatment. So, in this study, DCA is mainly an experimental tool to mimic chemical insults.
3. Figure 1b/1c shows some staining of Jag1 in the superficial layer. Is that true staining or due to marginal effect of staining? Jag1/2 knockout in p63+ cells needs to be validated by immunostaining.
4. Figure 1e/6d appears to show increased keratinization in Jag1/2KO esophagus and forestomach in comparison to WT esophagus. Please explain why.
5. 4NQO is known to produce squamous cell carcinoma in the esophagus and tongue as well as the forestomach. Phenotypes of the 4NQO-treated tongue and esophagus need to

be included.

6. Jag1/2KO exacerbates “chemically induced inflammation” instead of “inflammatory disease”.

Reviewer #2 (Remarks to the Author):

In this research, Huang et al. report that Jag1/2 expression in basal cells regulates esophageal homeostasis and suppresses tumorigenesis of foregut. P63Cre mouse was used as a genetic tool to delineate the role of Jag1/2 specifically in basal cells. Loss of Jag1/2 in p63-expressing cells resulted in increased proliferation of basal progenitor cells, defective differentiation of epithelial cells and loss of cell polarity of basal progenitor cells. In a mouse model of DCA-induced gastroesophageal inflammation, P63CreJag1/2f/f mice developed exacerbated disease accompanied with increased myeloid cell infiltration. In 4-NQO-induced SCC model, P63CreJag1/2f/f mice manifested worsened pathology, more weight loss and rate of mortality. Finally, reduced Jag2 expression significantly correlated with poor survival of ESCC patients. These findings are of potential interest. However, key weakness in this research was reflected in the following aspects. First, it is not appropriate to use WT mouse as a control for P63Cre-ERT2Jag1/2f/f mouse. According the reference described (PMID: 25210499), Cre-ERT2 was knocked into exon 4 of p63, disrupting one allele of p63. Therefore, this mouse functions as a p63 heterozygous mouse. Although it was not described clearly anywhere in the manuscript, I believe that the authors used P63Cre-ERT2/+Jag1/2f/f for their experiments (described as P63Cre-ERT2Jag1/2f/f). Please describe the genotype clearly in the manuscript and use P63Cre-ERT2/+Jag1/2+/+ mouse as a control for their experiments. This point is critical also because it has been shown previously that the basal cells are heterogeneous containing “p63+/K14-/K5- (3.2%), p63-/K14-/K5+ (5.6%), p63+/K14-/K5+ (4.0%), p63-/K14+/K5- (1.6%), p63+/K14+/K5- (9.6%), p63-/K14+/K5+ (14.6%) and p63+/K14+/K5+ (61.4%)” for the prostate basal cells (PMID: 25210499). This could affect the conclusions drawn from the observation in Fig. 3, as Krt5 could be differentially expressed by p63+ and p63- basal cells. Second, it is not known whether loss of Jag1 and Jag2 alone in p63+ cells is sufficient to cause abnormality of basal cells. Third, mechanism of how Jag1/2 affect the proliferation/differentiation of basal cells is not clearly addressed. As Jag1/2 are ligands of Notch, they are supposed to function through acting on Notch-expressing cells, further leading to changed expression of downstream target genes (such as Hes1, Integrins and ZO-1 etc).

Specific points:

1. In this research, the authors deleted Jag1 and Jag2 simultaneously in p63+ cells (including the progenies of p63+ cells). Was Jag1 and Jag2 functionally redundant in regulating basal cell homeostasis? How was the phenotype of p63Cre crossed to Jag1f/f or Jag2f/f mouse (loss of Jag1 or Jag2 alone in p63+ cells)?
2. Jag1 and Jag2 are ligands of Notch. Limited evidence supported that Notch signaling were severely impaired upon deletion of Jag1/2, except for reduced Hes1 expression. Jag1/2 were supposed to act on Notch-expressing cells. Upon deletion of Jag1/2, it remains unclear what cells are direct targets of Jag1/2. Deletion of ligands may affect all Notch-expressing cells in the microenvironment, including stroma cells, immune cells etc. In terms of findings of this research, it was unclear if Integrins, E-cadherin, ZO-1 were direct targets of Jag1/2-Notch/RBP-J signaling pathway. Does Jag1/2 expression in basal cells mainly affect Notch signaling within the basal cells themselves? In vitro experiments will be helpful to address this point.

3. Fig6: It is unclear if the weight loss/lethality of KO mice were due to squamous cell carcinoma. As p63Cre also delete basal cells in the lung, bladder, etc, Jag1/2 KO mice may also have abnormalities in other organs, which is important to look at.

4. Figure 4b-f: The authors showed a very small scope of immunofluorescence picture demonstrating the changed expression of Itga6, Itga1, Itgb4 and E-cadherin and PKC ζ , which was hard to reflect the real change of whole esophageal tissue. Please use a smaller magnification to depict a wider scope of the whole tissue. Also, please perform statistical analysis on the expression of Itga6, Itga1, Itgb4 and E-cadherin and PKC ζ . It will also be helpful to analyze the expression of Itga6, Itga1, Itgb4 and E-cadherin and PKC ζ using western blot with the protein lysates of esophageal epithelia, which more accurately represents the change of these molecules in the whole tissue.

Other points:

1. It was not clearly described where the Jag1/Jag2 mice were from and how they were the generated. Line 357.
2. RNA-seq data should be deposited in a public database.
3. Y axis label is missing for Figure. 4a.

Reviewer #3 (Remarks to the Author):

In the manuscript by Zhang and colleagues state-of-the-art genetic mouse models were used to define the role of Jag1/2 in esophageal homeostasis and tumor initiation. The manuscript is a strong continuation around the tumor suppressive function of Notch signaling in esophageal cancer. Loss of function of Jag1 and Jag2 describes the importance of this signaling axis for homeostatic conditions and the generation of self-renewing basal progenitors, drives inflammation and tumor initiation of the squamous epithelium of the forestomach. The experiments are sound and the manuscript is clearly written and presented. The major concern is the lack of mechanistic understanding of the observed phenotypes. Therefore, the current version of the manuscript is too descriptive for publication in Nature Communications.

Major comments:

Lacking mechanism

As mentioned, there are a number of interesting observations assembled, however, which mechanistic determinations are crucial for the observed stem cell dynamics, immune cell infiltration and tumor initiation is not clear. To gain causative understanding, at least one of these aspects should be dissected in much more mechanistic detail.

Human relevance

The presented analysis of human data in Fig.8 is relatively sparse and should be solidified, also along the lines of the mechanism's aspects. In addition, staining of JAG1 and JAG2 on human ESCC would be highly informative to understand spatial heterogeneity and potential effects of lateral Notch inhibition.

REVIEWER COMMENTS

Reviewer #1 (Remarks to the Author):

This manuscript by Huang et al carefully characterized the morphological and molecular phenotypes in Jag1/2KO mouse esophagus and showed certain relevance to human esophagitis and esophageal squamous cell carcinoma. This study is an extension of their published work on the role of Notch pathway in embryonic esophageal development (Ref#37). It is, in fact, the first one to systematically understand the role of Notch ligands (Jag1/2) in the homeostasis of adult esophageal epithelium and esophageal diseases. This reviewer has several major concerns and minor concerns as follows:

We appreciate the reviewer's constructive comments and, more importantly, the insightful suggestions. We have conducted the recommended experiments and incorporated the valuable suggestions, which have significantly helped strengthen the work, especially in the aspects of mechanistic studies and clinical relevance.

Major concerns:

1. The positional relationship between the ligands, receptors, and effectors of the Notch pathway is not clear. For example, immunofluorescent staining of Jag1, Notch1 (maybe NICD1), and Hes1 may address this issue.

We thank the reviewer's insightful suggestions and have accordingly conducted the recommended immunostaining. The immunostaining shows that Jag1/2 are mainly expressed in the basal cells of the esophageal epithelium (**Fig. 1b, c**). Notch1 localized to the membrane of basal cells, but mainly in the nucleus of suprabasal cells (**Fig. 5d**). Western blot analysis revealed a decrease in the protein levels of NICD1 and immunostaining showed a reduction in the number of nuclear Notch1⁺ suprabasal cells upon Jag1/2 deletion (**Fig. 5b, d**). Based on these analyses, we hypothesize that Jag1/2 on basal cells activate Notch signaling in suprabasal cells to promote their differentiation, while a deletion of Jag1/2 impairs Notch signaling, resulting in a reduction of epithelial differentiation. We also attempted immunostaining of Hes1 and NICD1 on tissues, but unfortunately, it was unsuccessful.

2. Clinical relevance of the findings needs to be clearly defined. For example, how are Jag1/2 and Dll1/3/4 expressed in human esophageal epithelium? Do they follow similar expression patterns as in the mouse esophageal epithelium?

We thank the reviewer's insightful questions. We analyzed RNA sequencing data from the Human Protein Atlas (HPA) database. Human esophagus expresses Notch ligands Jag1/2 and Dll1/3/4 in a similar pattern to the mice, with Jag1 showing the highest expression (**Supplementary Fig. 1d**).

Are there any differences in Jag1/2 expression in normal vs GERD esophageal epithelium? Does Jag1/2 downregulation take place in human esophageal squamous cell carcinoma in comparison with the normal esophageal epithelium? Is Jag1/2 downregulation an early or late event during esophageal carcinogenesis?

We thank the reviewer's insightful questions. To answer these questions, we have analyzed single-cell RNA sequencing data of human esophageal epithelial cells of normal, inflammation, neoplasia, and ESCC (**Fig. 9a, b**). First, we agreed with the reviewer's insight below that the DCA-treatment mouse model serves as an experimental tool to mimic chemical insults, which is characterized by inflammation.

As shown in **Fig. 9a, b**, both JAG1 and JAG2 exhibited lower expression in human esophageal epithelial cells under inflammatory conditions compared to the normal state. Moreover, there is also a downregulation of JAG1/2 in esophageal epithelial cells with neoplasia (**Fig. 9a, b**). However, JAG1/2 are unregulated in the ESCC compared to the normal esophageal epithelium, as evidenced by both single-cell RNA sequencing and immunostaining of human ESCC samples (**Fig. 9a-d**). These findings demonstrated that JAG1/2 downregulation is an early event in ESCC carcinogenesis. Our findings corroborate previous studies that have established Notch signaling as a tumor suppressor of squamous cell carcinoma (SCC) initiation¹⁻⁴. Nevertheless, contrasting evidence suggests a pro-tumorigenic role of Notch signaling during SCC progression^{5,6}. In line with this, we found that the expression levels of JAG1/2 were elevated in the human ESCC samples compared to normal samples. It is possible that Jag1/2-mediated Notch signaling exerts distinct functions at different stages of ESCC carcinogenesis, which is suppressing the onset of ESCC while promoting its progression. However, validating these hypotheses necessitates further investigation through comprehensive mouse genetic studies by modulating Jag1/2-Notch signaling at various stages of carcinogenesis. We have included these findings and details in the results (**lines 297-308**) and discussion sections (**lines 388-399**). Again, we appreciate the reviewer's insightful suggestions, which have significantly improved the clinical relevance of our studies.

One critical and important issue is whether the mouse models (Figure 5a/6a) recapitulate human diseases or just a biological phenomenon. "Deregulation of homeostasis", "perturbed epithelium", etc are biological terms, not terms in clinical pathology. A clinical pathologist may be consulted regarding these issues related to clinical.

The 4-NQO mouse model has been widely utilized in the field to investigate the mechanisms underlying ESCC carcinogenesis. As the reviewer suggested, we have consulted a pathologist and confirmed that the histological features observed in this model resemble those of human ESCC. Regarding the DCA-treatment mouse model, we agreed with the reviewer's assessment that it serves as an experimental model to mimic chemical insults. We have integrated this insight into the revised manuscript.

Overall, these mouse models bear significant clinical relevance in addition to their biological significance.

3. Lack of mechanistic in-depth is a concern. For example, how does Jag1/2KO impact the PI3K pathway?

We thank the reviewer's insightful questions. To elucidate the molecular mechanisms, we have conducted the following experiments. Given that Jag1/2 are important ligands for activating Notch signaling, which has been shown essential for esophageal epithelial cell differentiation^{7,8}, we set out to determine whether the ablation of Jag1/2 impairs the Notch signaling pathway to reduce cell differentiation. We first conducted gene set enrichment analysis (GSEA) on the RNA-sequencing data by comparing the esophageal epithelium of the *Jag1/2 cKO* mutants to the *control*. The analysis revealed a downregulation of Notch signaling in the *Jag1/2 cKO* mutants (**Fig. 5a**). Subsequent western blot analysis demonstrated a reduction in the protein levels of Notch1 intracellular domain (NICD1), the activated form of Notch1, upon Jag1/2 deletion (**Fig. 5b**). The expression of the differentiation marker *Krt4* was decreased (**Fig. 5c**), in line with the immunostaining analysis (**Fig. 3e, f**). Furthermore, immunostaining showed that the nuclear Notch1⁺ cells are reduced in the esophageal epithelium (**Fig. 5d, e**). These results demonstrated that Jag1/2 deletion impaired the activation of Notch signaling.

To further investigate whether Jag1/2 directly modulate epithelial cell differentiation, we generated 3D organoids from mouse esophageal epithelial cells (**Fig. 5f**). Deletion of Jag1/2 in the organoids led to the upregulation of the basal cell marker *p63* but a decrease in the expression of the suprabasal cell marker *Krt4* upon Jag1/2 deletion (**Fig. 5g**). Western blot analysis consistently showed a downregulation of *Krt4* protein levels (**Fig. 5h**). Moreover, immunostaining on organoids revealed an increase in p63⁺ basal cells while showing a reduction in squamous differentiation with Jag1/2 deletion (**Fig. 5i-k**). Moreover, the protein levels of NICD1 were downregulated (**Fig. 5h**), and nuclear Notch1⁺ cells were reduced in *Jag1/2 KO* organoids (**Fig. 5l**). Together, these results suggest that Jag1/2 within esophageal epithelial cells promote squamous differentiation through the Notch signaling pathway.

In terms of PI3KCG, PI3KCG has been identified as a direct target of Notch/Rbpjk signaling (PMID: 25808869)⁹. While the study revealed that Notch signaling enhanced PI3KCG expression in breast cancer, our findings showed reduced PI3KCG expression in *Jag1/2 cKO* mutants. This discrepancy likely arises because Notch signaling functions differently in a context-dependent manner. Additionally, we have also attempted to conduct Rbpjk and NICD1 ChIP-seq studies. Regrettably, the experiments did not yield conclusive results. It will be of significance to investigate whether this gene and other differentially expressed genes (DEGs) are direct targets of the Jag1/2-mediated Notch signaling pathway in the esophageal epithelium in future studies.

Overall, our mechanistic studies support that *Jag1/2 KO* reduces squamous epithelial differentiation, leading to basal cell hyperplasia, by impairing Notch signaling activation.

Why does *Jag1/2KO* cause body weight loss and death when the mice were given 4NQO (Figure 5b/6b)? Why does *Jag1/2KO* cause inflammation? If compromised epithelial barrier function and subsequent bacterial infection are important factors, additional experimental data need to be presented, for example, using germ-free mice.

We thank the reviewer's questions. Our initial studies showed that *Jag1/2 KO* leads to increased intercellular space in the foregut squamous epithelium, resulting in enhanced epithelial permeability upon insults (**Supplementary Fig. S3**). Both DCA and 4-NQO are potent squamous epithelial damaging agents, and the increased permeabilization of these agents in *Jag1/2 KO* mutants exacerbated injury, leading to increased inflammation (**Fig. 6d, h, i**). Additionally, the expression levels of pro-inflammatory cytokines were increased in the epithelial cells of *Jag1/2* mutants (**Supplementary Fig. 7**). Therefore, the enhanced inflammation likely resulted from a combined effect involving both chemical insult and the inflammatory cytokines. Moreover, both the control and *Jag1/2 KO* mutants were maintained under specific-pathogen-free conditions, and increased inflammation in *Jag1/2 KO* mutants was only observed following treatment with DCA or 4-NQO. Hence, subsequent infections are not assumed to be the primary factors for the in these models. That being said, it will be of significance to determine whether *Jag1/2 KO* mice are more susceptible to the pathogenic infection in future studies. This insight was incorporated into the discussion section (**lines 375-378**).

4. Dilated intercellular space and leaky epithelium are possible consequences of *Jag1/2KO*. TEM and TEER analysis are needed to validate these structural and functional phenotypes in the esophageal epithelium.

We thank the reviewer's insightful suggestion. We conducted the analyses as suggested. Transmission electron microscope (TEM) analysis showed a consistent dilated intercellular space between basal cells in *Jag1/2 cKO* mutants (**Fig. 4f**). Regarding the functional analysis, performing TEER measurements on mouse foregut mucosa is highly challenging. Thus, we turned to Evans blue dye staining as an alternative method^{10,11}. The staining revealed increased permeability in the *Jag1/2 cKO* forestomach mucosa treated with DCA (**Supplementary Fig. 3**).

Minor concerns:

1. Protein names and gene names need to be expressed consistently and follow the requirements of the journal. For example, most cancer journals may require protein names to be capitalized.

We thank the reviewer's comments. The spelling of the protein and gene names has been corrected. To clarify, the first letter was capitalized for mouse protein and gene names, while all letters were capitalized for human protein and gene names.

2. DCA should be deoxycholic acid instead of dichloroacetate. DCA is rarely detected in the upper gastrointestinal tract in humans. Treatment with gastric content (for example, acid, enzymes, conjugated bile acids) is more clinically relevant than DCA treatment. So, in this study, DCA is mainly an experimental tool to mimic chemical insults.

We appreciate the reviewer's corrections and insightful suggestions. We have corrected the spelling to "deoxycholic acid". We have incorporated the reviewer's insight that DCA treatment is mainly utilized as an experimental tool to mimic chemical insults.

3. Figure 1b/1c shows some staining of Jag1 in the superficial layer. Is that true staining or due to marginal effect of staining? Jag1/2 knockout in p63+ cells needs to be validated by immunostaining.

We thank the reviewer's questions. The staining on the superficial layers is marginal staining, and we have added annotations to clarify in the figure legends (**Fig. 1b, c and Supplementary Fig. 1c, d**). The immunostaining confirming the Jag1/2 knockout in the esophageal epithelium in *Jag1/2 cKO* mutants was included (**Supplementary Fig. 1e, f**).

4. Figure 1e/6d appears to show increased keratinization in Jag1/2KO esophagus and forestomach in comparison to WT esophagus. Please explain why.

We thank the reviewer's questions. Original Figure 1e: *WT* and *Jag1/2KO* esophagus displayed similar keratinization. We have re-cropped the representative enlarged image for *WT* for clarification. Original Figure 6d: large tumors developed in the *Jag1/2 KO* mutants, with tumor cells expanding towards the lumen and undergoing apoptosis and eventually keratinization. Therefore, the increased keratinization is a result of the expansion of tumor cells in the mutants.

5. 4NQO is known to produce squamous cell carcinoma in the esophagus and tongue as well as the forestomach. Phenotypes of the 4NQO-treated tongue and esophagus need to be included.

We thank the reviewer's suggestions. We conducted histological analysis on the tongue and esophagus treated with 4NQO. *Jag1/2 cKO* mice showed more severe dysplasia compared to the control mice (**Supplementary Fig. 6a, b**).

6. Jag1/2KO exacerbates “chemically induced inflammation” instead of “inflammatory disease”.

We thank the reviewer’s suggestion. We have corrected the terminology.

Reviewer #2 (Remarks to the Author):

In this research, Huang et al. report that Jag1/2 expression in basal cells regulates esophageal homeostasis and suppresses tumorigenesis of foregut. P63Cre mouse was used as a genetic tool to delineate the role of Jag1/2 specifically in basal cells. Loss of Jag1/2 in p63-expressing cells resulted in increased proliferation of basal progenitor cells, defective differentiation of epithelial cells and loss of cell polarity of basal progenitor cells. In a mouse model of DCA-induced gastroesophageal inflammation, P63CreJag1/2f/f mice developed exacerbated disease accompanied with increased myeloid cell infiltration. In 4-NQO-induced SCC model, P63CreJag1/2f/f mice manifested worsened pathology, more weight loss and rate of mortality. Finally, reduced Jag2 expression significantly correlated with poor survival of ESCC patients. These findings are of potential interest. However, key weakness in this research was reflected in the following aspects.

We appreciate the reviewer’s comments and the valuable suggestions. We have conducted the recommended experiments and incorporated the insightful suggestions, which have significantly helped strengthen our studies. A point-to-point response was provided as below.

First, it is not appropriate to use WT mouse as a control for P63Cre-ERT2Jag1/2f/f mouse. According the reference described (PMID: 25210499), Cre-ERT2 was knocked into exon 4 of p63, disrupting one allele of p63. Therefore, this mouse functions as a p63 heterozygous mouse. Although it was not described clearly anywhere in the manuscript, I believe that the authors used P63Cre-ERT2/+Jag1/2f/f for their experiments (described as P63Cre-ERT2Jag1/2f/f). Please describe the genotype clearly in the manuscript and use P63Cre-ERT2/+Jag1/2+/+ mouse as a control for their experiments. This point is critical also because it has been shown previously that the basal cells are heterogeneous containing “p63+/K14-/K5- (3.2%), p63-/K14-/K5+ (5.6%), p63+/K14-/K5+ (4.0%), p63-/K14+/K5- (1.6%), p63+/K14+/K5- (9.6%), p63-/K14+/K5+ (14.6%) and p63+/K14+/K5+ (61.4%)” for the prostate basal cells (PMID: 25210499). This could affect the conclusions drawn from the observation in Fig. 3, as Krt5 could be differentially expressed by p63+ and p63- basal cells.

We appreciate the reviewer's insightful suggestions. We have changed the spelling of “*p63^{CreERT2}*” to “*p63^{CreERT2/+}*” for clarity. As advised, we have further conducted homeostatic experiments utilizing *p63^{CreERT2/+}* mice. These mice retained the expression of p63 and the esophageal tissues showed no phenotypical changes compared to *WT* mice during homeostasis (Figs. 1e, 3b, e and Supplementary Fig. 1i-

k). Additionally, we also treated *p63^{CreERT2/+}* mice with 4-NQO in the SCC studies, and these mice displayed similar phenotypical changes to the *WT* mice (**Fig. 7c-e and Supplementary Fig. 4b-d**). Therefore, these findings consistently support the conclusion that the *Jag1/2* deletion disrupted squamous epithelial homeostasis and accelerated tumor initiation.

Second, it is not known whether loss of Jag1 and Jag2 alone in p63+ cells is sufficient to cause abnormality of basal cells.

We are grateful for the reviewer's insightful question. We have conducted the suggested studies by individually deleting *Jag1* or *Jag2* alone. A detailed reply was provided in the response to “Specific point 1”.

Third, mechanism of how Jag1/2 affect the proliferation/differentiation of basal cells is not clearly addressed. As Jag1/2 are ligands of Notch, they are supposed to function through acting on Notch-expressing cells, further leading to changed expression of downstream target genes (such as Hes1, Integrins and ZO-1 etc).

We are grateful for the reviewer's insightful question. We have conducted experiments using various molecular and cellular approaches to address the question. A detailed reply was provided in the response to “Specific point 2”.

Specific points:

1. In this research, the authors deleted Jag1 and Jag2 simultaneously in p63+ cells (including the progenies of p63+ cells). Was Jag1 and Jag2 functionally redundant in regulating basal cell homeostasis? How was the phenotype of p63Cre crossed to Jag1f/f or Jag2f/f mouse (loss of Jag1 or Jag2 alone in p63+ cells)?

We thank the reviewer's question. The esophageal epithelium of *Jag1 KO* or *Jag2 KO* mice also display a similar phenotype to the *Jag1/2 cKO* mutants, albeit with less severities (**Fig. 1e and Supplementary Fig. 1g, h**) suggesting a redundant role of these two ligands. Therefore, the studies focused on the *Jag1/2 KO* mutants.

2. Jag1 and Jag2 are ligands of Notch. Limited evidence supported that Notch signaling were severely impaired upon deletion of Jag1/2, except for reduced Hes1 expression. Jag1/2 were supposed to act on Notch-expressing cells. Upon deletion of Jag1/2, it remains unclear what cells are direct targets of Jag1/2. Deletion of ligands may affect all Notch-expressing cells in the microenvironment, including stroma cells, immune cells etc. In terms of findings of this research, it was unclear if Integrins, E-cadherin, ZO-1 were direct targets of Jag1/2-Notch/RBP-J signaling pathway. Does Jag1/2 expression in basal cells mainly affect Notch signaling within the basal cells themselves? In vitro experiments will be helpful to address this point.

We thank the reviewer's insightful questions. Given that Jag1/2 are important ligands for activating Notch signaling, which has been shown essential for esophageal epithelial cell differentiation^{7,8}, we determined whether the ablation of Jag1/2 impairs the Notch signaling pathway to reduce cell differentiation. We first conducted gene set enrichment analysis (GSEA) on the RNA-sequencing data by comparing the esophageal epithelium of the *Jag1/2 cKO* mutants to the *control*. The analysis revealed a downregulation of Notch signaling in the *Jag1/2 cKO* mutants (**Fig. 5a**). Subsequent western blot analysis demonstrated a reduction in the protein levels of Notch1 intracellular domain (NICD1), the activated form of Notch1, upon Jag1/2 deletion (**Fig. 5b**). The expression of differentiation marker *Krt4* was decreased (**Fig. 5c**), in line with the immunostaining analysis (**Fig. 3e, f**). Furthermore, immunostaining showed that the nuclear Notch1⁺ cells are reduced in the esophageal epithelium (**Fig. 5d, e**). These results demonstrated that the Jag1/2 deletion impaired the activation of Notch signaling. Previous studies conducted by our group and others have shown that Notch signaling is required for esophageal epithelial cell differentiation^{7,8}. Therefore, these findings demonstrated that the Jag1/2 ablation impaired Notch signaling leading to reduced cell differentiation.

To further investigate whether Jag1/2 directly modulate epithelial cell differentiation, we generated 3D organoids from mouse esophageal epithelial cells (**Fig. 5f**). Deletion of Jag1/2 in the organoids led to the upregulation of the basal cell marker *p63* but a decrease in the expression of the suprabasal cell marker *Krt4* upon Jag1/2 deletion (**Fig. 5g**). Western blot analysis consistently showed a downregulation of *Krt4* protein levels (**Fig. 5h**). Moreover, immunostaining on organoids revealed an increase in *p63*⁺ basal cells while showing a reduction in squamous differentiation with Jag1/2 deletion (**Fig. 5i-k**). Moreover, the protein levels of NICD1 were downregulated (**Fig. 5h**), and nuclear Notch1⁺ cells were reduced in *Jag1/2 KO* organoids (**Fig. 5l**). These results together suggest that Jag1/2 within esophageal epithelial cells promote squamous differentiation through the Notch signaling pathway.

We have also endeavored to conduct Rbpjk and NICD1 ChIP-seq studies to determine whether Integrins, E-cadherin, ZO-1 are direct targets of the Jag1/2-Notch/RBP-J signaling pathway. Regrettably, the experiments did not yield conclusive results. Notably, previous studies have demonstrated that the Notch signaling pathway negatively regulates the expression of the basal cell master transcription factor *p63*, leading to cell differentiation in skin keratinocytes¹². Additionally, *p63* has been shown to directly transactivate the expression of *Itga6*, *Itgb1*, and *Itgb4*¹³. Our findings revealed an increase in the expression levels of *p63*, and expansion of *p63*⁺ basal cells with Jag1/2 deletion (**Figs. 3b-d, 5g, i, j**). It is possible that the reduced epithelial differentiation and increased expression of integrins in the *Jag1/2 KO* mutants are attributed to the upregulation of *p63*. We have included this potential explanation in the discussion section (**lines 360-366**). Overall, our studies elucidated a mechanism that Jag1/2 deficiency diminishes esophageal epithelial differentiation through the Notch signaling pathway.

3. Fig6: It is unclear if the weight loss/lethality of KO mice were due to squamous cell carcinoma. As p63Cre also delete basal cells in the lung, bladder, etc, Jag1/2 KO mice may also have abnormalities in other organs, which is important to look at.

We thank the reviewer's insightful suggestions. We have conducted the suggested studies. *Jag1/2 KO* mice developed severe SCC in the forestomach at around 4-5 weeks (**Fig. 6b-c**). Once tumors developed, *Jag1/2 KO* mice exhibited illness, ceased food intake, and eventually succumbed (**Supplementary Fig. 4a**). Histological analyses showed that there were no apparent phenotypic changes in the lung and bladder of both the *Jag1/2 cKO* mutants and the control group (**Supplementary Fig. 6c, d**). Based on these analyses, we reasoned that mouse mortality is majorly attributed to the tumors developing in the forestomach.

4. Figure 4b-f: The authors showed a very small scope of immunofluorescence picture demonstrating the changed expression of Itga6, Itga1, Itgb4 and E-cadherin and PKC ζ , which was hard to reflect the real change of whole esophageal tissue. Please use a smaller magnification to depict a wider scope of the whole tissue. Also, please perform statistical analysis on the expression of Itga6, Itga1, Itgb4 and E-cadherin and PKC ζ . It will also be helpful to analyze the expression of Itga6, Itga1, Itgb4 and E-cadherin and PKC ζ using western blot with the protein lysates of esophageal epithelia, which more accurately represents the change of these molecules in the whole tissue.

We appreciate the reviewer's suggestions. As suggested, we included large images for all staining of Itga6, Itgb1, Itgb4 and E-cadherin and PKC ζ (**Fig. 4b-e, g**). Quantification of fluorescence intensity, showing the upregulation of the protein Itga6, Itgb1, and Itgb4 proteins, has also been included (**Fig. 4b-d**). Additionally, we attempted western blot analysis using antibodies for Itga6, Itgb1, and Itgb4 from at least two sources, but regrettably, none yielded successful results. Furthermore, the immunostaining of E-cadherin and PKC ζ was conducted to illustrate a dilated intercellular space and disrupted basal cell orientation, respectively. Their expression levels were not changed, and therefore quantification of their fluorescence intensity was not included. Overall, we are grateful for the reviewer's suggestions, which helped enhance the clarity of the data presentation.

Other points:

1. It was not clearly described where the Jag1/Jag2 mice were from and how they were generated. Line 357.

The references have been added accordingly.

2. RNA-seq data should be deposited in a public database.

The RNA-seq data have been deposited in the Genome Sequence Archive (GSA) database under accession number CRA011089 and can be accessed for review through the following link: <https://ngdc.cnpc.ac.cn/gsa/s/UDe7kx7h>.

3. Y axis label is missing for Figure. 4a.

We have added the Y axis label accordingly.

Reviewer #3 (Remarks to the Author):

In the manuscript by Zhang and colleagues state-of-the-art genetic mouse models were used to define the role of Jag1/2 in esophageal homeostasis and tumor initiation. The manuscript is a strong continuation around the tumor suppressive function of Notch signaling in esophageal cancer. Loss of function of Jag1 and Jag2 describes the importance of this signaling axis for homeostatic conditions and the generation of self-renewing basal progenitors, drives inflammation and tumor initiation of the squamous epithelium of the forestomach. The experiments are sound and the manuscript is clearly written and presented. The major concern is the lack of mechanistic understanding of the observed phenotypes. Therefore, the current version of the manuscript is too descriptive for publication in Nature Communications.

We appreciate the reviewer's constructive comments. To strengthen the mechanistic studies, following the suggestions, we have employed esophageal-derived 3D organoids and biochemical approaches to demonstrate that Jag1/2 directly regulates foregut epithelial differentiation by activating Notch signaling. Furthermore, we also follow the reviewer's suggestion to look into the expression of Jag1/2 in human ESCC using single cell-seq data and immunostaining of human ESCC tissue arrays. Significantly, we have found that Jag1/2 are down-regulated at the early stages of tumor initiation, which are inflammation and neoplasia, while their expression is increased in the tumors. These data suggest a distinct role of Jag1/2 in ESCC tumor initiation and progression. A detailed response to these questions well described as below.

Major comments:

Lacking mechanism

As mentioned, there are a number of interesting observations assembled, however, which mechanistic determinations are crucial for the observed stem cell dynamics, immune cell infiltration and tumor initiation is not clear. To gain causative understanding, at least one of these aspects should be dissected in much more mechanistic detail.

We thank the reviewer's insightful questions. As suggested, we chose to further investigate the mechanisms by focusing on the stem cell dynamics utilizing 3D organoids and biochemical approaches.

Given that Jag1/2 are important ligands for activating Notch signaling which has been shown essential for esophageal epithelial cell differentiation^{7,8}, we set out to determine whether the ablation of Jag1/2 impairs the Notch signaling pathway to reduce cell differentiation. We first conducted gene set enrichment analysis (GSEA) on the RNA-

sequencing data comparing the esophageal epithelium of the *Jag1/2 cKO* mutants to the *control*. The analysis revealed a downregulation of Notch signaling in the *Jag1/2 cKO* mutants (**Fig. 5a**). Subsequent western blot analysis demonstrated a reduction in the protein levels of Notch1 intracellular domain (NICD1), the activated form of Notch1, upon *Jag1/2* deletion (**Fig. 5b**). The expression of the differentiation marker *Krt4* was decreased (**Fig. 5c**), in line with the immunostaining analysis (**Fig. 3e, f**). Furthermore, immunostaining showed that the nuclear Notch1⁺ cells are reduced in the esophageal epithelium (**Fig. 5d, e**). These results demonstrated that *Jag1/2* deletion impaired the activation of Notch signaling.

To further investigate whether *Jag1/2* directly modulate epithelial cell differentiation, we generated 3D organoids from mouse esophageal epithelial cells (**Fig. 5f**). Deletion of *Jag1/2* in the organoids led to the upregulation of the basal cell marker *p63* but a decrease in the expression of the suprabasal cell marker *Krt4* upon *Jag1/2* deletion (**Fig. 5g**). Western blot analysis consistently showed a downregulation of *Krt4* protein levels (**Fig. 5h**). Moreover, immunostaining on organoids revealed an increase in *p63*⁺ basal cells while showing a reduction in squamous differentiation with *Jag1/2* deletion (**Fig. 5i-k**). Moreover, the protein levels of NICD1 were downregulated (**Fig. 5h**), and nuclear Notch1⁺ cells were reduced in *Jag1/2 KO* organoids (**Fig. 5l**). These results together suggest that *Jag1/2* within esophageal epithelial cells promote squamous differentiation through the Notch signaling pathway.

Overall, our mechanistic studies support that *Jag1/2 KO* reduces squamous epithelial differentiation, leading to basal cell hyperplasia, by decreasing Notch signaling. This in line with the overall conclusion that the disruption of *Jag1/2*-mediated Notch signaling reduces the differentiation of the basal cells, leading to expansion of basal progenitor cells and facilitating SCC initiation. We thank the reviewer's suggestion which have helped strengthen the mechanisms of the study.

Human relevance

The presented analysis of human data in Fig.8 is relatively sparse and should be solidified, also along the lines of the mechanism's aspects. In addition, staining of JAG1 and JAG2 on human ESCC would be highly informative to understand spatial heterogeneity and potential effects of lateral Notch inhibition.

We thank the reviewer's insightful questions. To answer these questions, we first analyzed the single-cell RNA sequencing data of the esophageal epithelial cells from four developmental stages of ESCC: normal, inflammatory, neoplastic, and ESCC, from available datasets¹⁴⁻¹⁷. The expression levels of both JAG1 and JAG2 were significantly lower at the inflammatory and neoplastic stages, which represent the early stages of ESCC carcinogenesis, compared to the normal state (**Fig. 9a, b**). However, their expression was higher in the ESCC samples compared to the normal stage (**Fig. 9a, b**). Consistently, immunostaining showed higher protein levels of both JAG1 and JAG2 in the human ESCC samples (**Fig. 9c, d and Supplementary Fig. 8a, b**). These

results demonstrate that JAG1/2 downregulation is an early event during ESCC carcinogenesis. Our findings corroborate previous studies that have established Notch signaling as a tumor suppressor of squamous cell carcinoma (SCC) initiation¹⁻⁴. Nevertheless, contrasting evidence suggests a pro-tumorigenic role of Notch signaling during SCC progression^{5,6}. In line with this, we found that the expression levels of JAG1/2 were elevated in the human ESCC samples compared to normal samples. It is possible that Jag1/2-mediated Notch signaling exerts distinct functions at different stages of ESCC carcinogenesis, which is suppressing the onset of ESCC while promoting its progression. However, validating these hypotheses necessitates further investigation through comprehensive mouse genetic studies by modulating Jag1/2-Notch signaling at various stages of carcinogenesis. We have included these findings and information in the results **(lines 297-308)** and discussion sections **(lines 388-399)**. Again, we appreciate the reviewer's insightful suggestions, which have significantly improved the clinical relevance of our studies.

References

1. Agrawal N, *et al.* Comparative genomic analysis of esophageal adenocarcinoma and squamous cell carcinoma. *Cancer Discov* **2**, 899-905 (2012).
2. Song Y, *et al.* Identification of genomic alterations in oesophageal squamous cell cancer. *Nature* **509**, 91-95 (2014).
3. Alcolea MP, Greulich P, Wabik A, Frede J, Simons BD, Jones PH. Differentiation imbalance in single oesophageal progenitor cells causes clonal immortalization and field change. *Nat Cell Biol* **16**, 615-622 (2014).
4. Sawangarun W, *et al.* Loss of Notch1 predisposes oro-esophageal epithelium to tumorigenesis. *Exp Cell Res* **372**, 129-140 (2018).
5. Natsuzaka M, *et al.* Interplay between Notch1 and Notch3 promotes EMT and tumor initiation in squamous cell carcinoma. *Nat Commun* **8**, 1758 (2017).
6. Lubin DJ, Mick R, Shroff SG, Stashek K, Furth EE. The notch pathway is activated in neoplastic progression in esophageal squamous cell carcinoma. *Hum Pathol* **72**, 66-70 (2018).
7. Ohashi S, *et al.* NOTCH1 and NOTCH3 coordinate esophageal squamous differentiation through a CSL-dependent transcriptional network. *Gastroenterology* **139**, 2113-2123 (2010).
8. Zhang Y, *et al.* 3D Modeling of Esophageal Development using Human PSC-Derived Basal Progenitors Reveals a Critical Role for Notch Signaling. *Cell Stem Cell* **23**, 516-529 e515 (2018).
9. Zhang S, Chung WC, Wu G, Egan SE, Miele L, Xu K. Manic fringe promotes a claudin-low breast cancer phenotype through notch-mediated PIK3CG induction. *Cancer Res* **75**, 1936-1943 (2015).
10. Di Simone MP, *et al.* Barrier effect of Esoxx((R)) on esophageal mucosal damage: experimental study on ex-vivo swine model. *Clin Exp Gastroenterol* **5**, 103-107 (2012).
11. Pecora TMG, *et al.* Barrier effect and wound healing activity of the medical device REF-FTP78 in the treatment of gastroesophageal reflux disease. *Sci Rep* **12**, 6136

- (2022).
12. Nguyen BC, *et al.* Cross-regulation between Notch and p63 in keratinocyte commitment to differentiation. *Genes Dev* **20**, 1028-1042 (2006).
 13. Carroll DK, *et al.* p63 regulates an adhesion programme and cell survival in epithelial cells. *Nat Cell Biol* **8**, 551-561 (2006).
 14. Zhang X, *et al.* Dissecting esophageal squamous-cell carcinoma ecosystem by single-cell transcriptomic analysis. *Nat Commun* **12**, 5291 (2021).
 15. Pan X, *et al.* Identifying a confused cell identity for esophageal squamous cell carcinoma. *Signal Transduct Target Ther* **7**, 122 (2022).
 16. Liu T, *et al.* Computational Identification of Preneoplastic Cells Displaying High Stemness and Risk of Cancer Progression. *Cancer Res* **82**, 2520-2537 (2022).
 17. Rochman M, *et al.* Single-cell RNA-Seq of human esophageal epithelium in homeostasis and allergic inflammation. *JCI Insight* **7**, (2022).

REVIEWERS' COMMENTS

Reviewer #1 (Remarks to the Author):

All my concerns have been addressed adequately.

Reviewer #2 (Remarks to the Author):

With additional experiments performed by the authors, the revised manuscript has been significantly improved. I have no further comments.

Reviewer #3 (Remarks to the Author):

The authors have addressed all the raised comments. Hence, the paper is suitable for publication.